# Spatial single-cell atlas reveals regional variations in healthy and diseased human lung

Alexandra B. Firsova [1,2], Sergio Marco Salas [1,3], Louis B. Kuemmerle[4,5], Xesús M. Abalo [1,6], Alexandros Sountoulidis [1,2], Ludvig Larsson[1,6], Krishnaa T. Mahbubani [7], Jonas Theelke [1,2], Zaneta Andrusivova[1,6], Leire Alonso Galicia[1,6], Andreas Liontos[1,2], Tamás Balassa[8], Ferenc Kovacs [8,9], Peter Horvath[8,9,10,11], Yuexin Chen[12,13], Janine Gote-Schniering [14,15], Mircea-Gabriel Stoleriu[13,16,17,18], Jürgen Behr[13,18,19], Kerstin B. Meyer [20], Wim Timens [21,22], Herbert B. Schiller[12,13,23], Malte D. Luecken [4,13,18], Fabian J. Theis [4,5,24], Joakim Lundeberg [1,6], Mats Nilsson [1,3], Martijn C. Nawijn [21,22] & Christos Samakovlis [1,2,25] ✉

Integration of scRNA-seq data from millions of cells revealed a high diversity of cell types in the healthy and diseased human lung. In a large and complex organ, constantly exposed to external agents, it is crucial to understand the influence of lung tissue topography or external factors on gene expression variability within cell types. Here, we apply three spatial transcriptomics approaches, to: (i) localize the majority of lung cell types, including rare epithelial cells within the tissue topography, (ii) describe consistent anatomical and regional gene expression variability within and across cell types, and (iii) reveal distinct cellular neighborhoods in specific anatomical regions and examine gene expression variations in them. We thus provide a spatially resolved tissue reference atlas in three representative regions of the healthy human lung. We further demonstrate its utility by defining previously unknown imbalances of epithelial cell type compositions in chronic obstructive pulmonary disease lungs. Our topographic atlas enables a precise description of characteristic regional cellular responses upon experimental perturbations or during disease progression.

Recent advances in single-cell omics have led to extensive reference datasets of cell types from various human organs, including the lung, harboring the respiratory system with its multitude of cell types and states[1–5]. Single-cell mRNA sequencing (scRNA-seq) unveiled previously unknown cell types in healthy human lung, such as ionocytes, hillock-like and tuft cells, neuroendocrine (NE) cell states and aerocytes[3,4,6–8]. Multiple subtypes of fibroblasts, immune and endothelial cells were also characterized based on their gene expression programs and inferred tissue distribution[4]. A limitation of these datasets deriving from dissociated tissue is that cell types and their annotation often lack the information on cellular location relative to tissue landmarks or along relevant axes in the tissue. For example, the proximo-distal axis in lung with gradients in, for instance, oxygen tension, is likely to impact on gene expression. Moreover, predicted intercellular interactions are based on selectivity of ligand/receptor pair expression at the mRNA level in cell-type pairs, but lack information about spatial proximity which is especially relevant at higher resolutions of cell subset annotation. On the other hand, creation of

spatially resolved gene expression maps can highlight physical proximity of cells and establish consistent cellular neighborhoods allowing focused analysis of cell-cell interactions. Such topographic atlases of healthy tissues in comparison to corresponding maps of diseased tissues can reveal potentially causative alterations in the cellular landscape in diseased organs.

The large size of the adult human lung precludes in-scale mapping of the full tissue. Instead, available atlases of the adult human lung rely on sampling of distinct anatomic regions from different donors[1–4,6,9–14]. For example, extensive proximal airway epithelial sampling allowed identification of nasal cell types, including nasal-specific serous, goblet and club cells[3]. Thorough characterization of distal airways focused on bronchial secretory cell populations, and defined the ATO intermediate cell state, characterized by the co-expression of bronchial secretory and alveolar epithelial Type 2 (AT2) markers[9]. A recent integration of scRNA-seq data has created a comprehensive cellular catalog of the healthy human lung, counting a total of 61 major cell clusters, 58 of those from the trachea and the lung. This defined 35 major cell types with 51 subtypes and seven cell states[2]. Most subtypes were annotated by sampling location and others by differential expression of single or few distinguishing markers. Cell states of different cell types were further classified as resting, proliferating, activated, or intermediate, suggesting the presence of cells in distinct steps of differentiation progression. The potentially distinct cellular environments of such transition states remain unknown.

Here, we generated a representative topographic atlas of the healthy adult lung by combining sections from distinct anatomic locations of the respiratory system from four donors of different age and gender. We used three different, multiplexed spatially-resolved transcriptomic (SRT) approaches to obtain complementary results. This included two targeted imaging-based methods with cellular resolution: a highly-multiplex HybISS[15] in order to target the majority of cell types, and a highly-sensitive SCRINSHOT[16] in order to detect a more limited number of cell types and states by variations in gene expression levels. We also employed an unbiased method of mRNA detection with lower spatial resolution[17] to confirm cell types and regional gene expression patterns (Fig. 1A). This combinatorial approach allowed us to deeply characterize consistent, location-related gene expression variability within and across cell types. Finally, we utilized diseased samples available to the consortium from patients diagnosed with chronic obstructive pulmonary disease (COPD), a serious lung disease with globally increasing incidence. This disease is characterized by damage of the respiratory system, such as small airways and alveoli, due to the long-term exposure to toxic particles and gases[18]. COPD samples provide an opportunity to investigate the utility of the healthy atlas in defining aberrant disease-associated cellular neighborhoods, affected by the immune cell composition of the lung, as well as tissue remodeling. We used the topographic atlas as a reference to define changes in cell-type and cell-state abundance and their spatial distribution in distal lung samples. We detected both cellular and neighborhood changes in the samples from three stage-II COPD patients revealing distinct cellular niches at an early stage of disease progression.

## Results

### A HybISS-based cell type map reveals specific cellular neighborhoods

We collected tissue samples from six donors targeting five discrete anatomical regions, congruent to the previously described locations of cells in scRNA-seq datasets[1,19], and grouped them into three major anatomical regions: trachea (ventral side of the airway with surrounding mesenchyme), proximal lung (generation 2-3 intralobar bronchus with surrounding mesenchyme and occasionally alveoli) and distal lung (distal/terminal bronchioli and alveolar tissue close to the edge of the lobes). After histology-based assessment, two out of six

donors were excluded due to multiple signs of pathology, including fibrosis or large immune infiltrations. Samples from the remaining four donors, among which two were smokers, one ex-smoker and one non-smoker (Supplementary Data 1), were subjected to mRNA quality controls to reject the samples with low or diffuse RNA signal (Methods, Supplementary Fig. 1A). High-quality samples from different locations were processed by three different complementary SRT technologies (Fig. 1A, Supplementary Table 1). In order to detect all cell types simultaneously on each tissue section, we first classified cells based on previously published scRNA-seq data (Supplementary Fig. 1B)[1,3] and generated a probe panel for HybISS using Spapros, as previously described[20]. We applied this panel on two selected representative samples using HybISS[15]. The obtained dataset was used to detect most cell types and states in the trachea and lung. In order to detect particular cell states, we used SCRINSHOT, a more sensitive SRT method, with an additional panel of major cell type markers and genes showing intra-cluster gene expression variation[16,20]. Finally, a section from each anatomic location was analyzed with an untargeted SRT method Visium in order to validate the probe panels. Visual cross-validation sections of the three methods on serial tissue sections demonstrated consistent marker gene expression patterns, and the performance of targeted methods (Supplementary Fig. 1C).

First, we identified cell types by profiling sections from three anatomic locations of four donors by highly multiplex method HybISS using a gene panel consisting of 162 genes (Supplementary Data 2). After decoding, 157 genes were annotated. Cells were segmented based on DAPI-stained nuclei, and gene transcript signals were assigned to the nearest nuclei[21]. We excluded cells with low transcript counts and finally processed a total of 260,398 cells for further analysis and clustering based on their expression profiles. This separated the cells into six major classes, assigned according to marker gene expression: airway epithelial, immune, alveolar epithelial, endothelial, stromal, and submucosal gland (SMG) (Supplementary Data 2, 3). The cells in these classes mapped to their expected histological locations (Supplementary Fig. 2A). By further subclustering of each class, we revealed and annotated 28 cell types (Fig. 1B, Supplementary Data 3), corresponding to the majority of the adult lung cell types, described in previous scRNA-seq studies[2,4]. Based on positivity for corresponding cell type marker genes in the RNA-seq atlases[1–3], we manually annotated seven additional cell types that could not be assigned by the unsupervised sub-clustering of the HybISS data either due to their sparsely detected gene expression, such as in T lymphocytes, NK cells, a group of T and/or NK cells (here labeled T/NK), and aerocytes, or due to their low abundance, such as ionocytes, tuft cells, rare tuft-like cells, and squamous-like cells (Supplementary Fig. 2B, Supplementary Table 2). The expected marker genes were expressed in the corresponding cell types, except for 11 genes, which were detected in very few cells of the annotated clusters (Supplementary Fig. 2C-D, in brackets). The overall performance of the HybISS marker gene probe panel and cell type annotations were tested using integration mapping[22] with the corresponding scRNA-seq dataset[1]. Cell types detected in HybISS demonstrated high prediction score corresponding to the expected scRNA-seq cell type annotations (Supplementary Fig. 3A). Ten cell types demonstrated prediction scores to more than one annotation, due to their mixed origin and marker co-expression with related types, or due to lack of cell type-specific markers. These cell types were annotated based on their location and morphology on histological images (Supplementary Fig. 3B). Their annotation and location was confirmed in complementary SCRINSHOT- and Visium-processed serial sections, as described below. In total, our analysis resulted in identification of a total of 35 cell types among 221,130 cells that were mapped onto the tissue topography in situ (Fig. 1B-D). The data were deposited in an open access searchable browser that visualizes cell type annotation, gene expression levels and histological stainings (see Data Availability in viewers for HybISS Atlas).

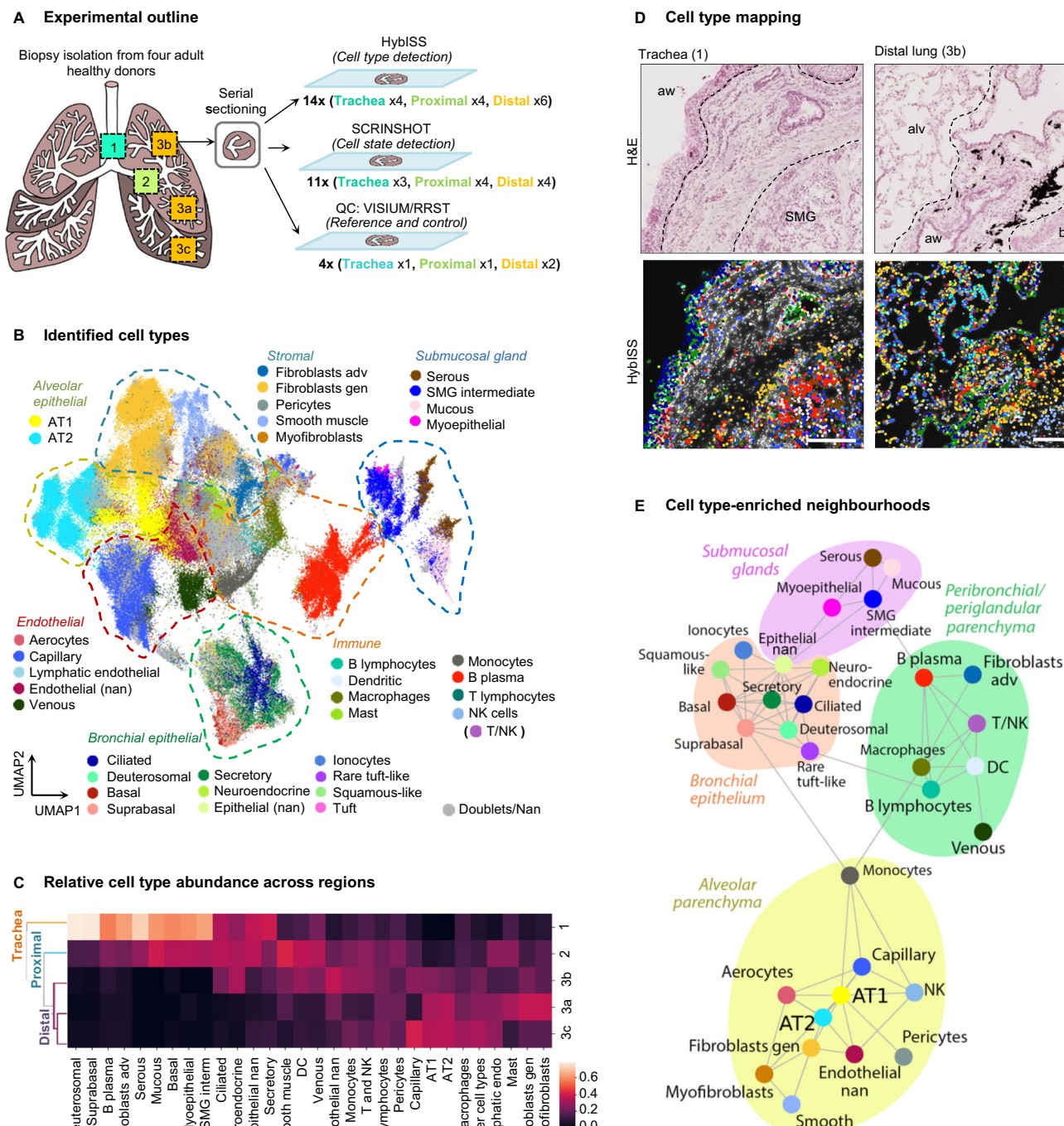

**Fig. 1 | HybISS-based cell type map reveals cell type distribution and neighborhoods. A** Experimental outline including the location of sample collection from donated healthy human lungs, and the methods used for the mRNA-based cell type mapping. 1−trachea, 2−proximal bronchi, 3a−bottom part of upper left lobe, 3b−top part of upper left lobe, 3c−bottom part of lower left lobe. HybISS Hybridization-based in situ sequencing, SCRINSHOT Single-Cell Resolution IN Situ Hybridization On Tissues, RRST RNA-Rescue Spatial Transcriptomics. **B** UMAP of cells after leiden clustering profiled with HybISS, colored by the assigned cell type (35 cell types presented). Gen general, adv adventitial, nan not annotated. **C** Heatmap of relative abundance of clustered cell types between locations demonstrating their frequency across the profiled regions. Similarity in cell type composition between three distal lung regions can be assessed by hierarchical clustering dendrogram on the left side. **D** Representative histological images from four biological replicates of an analysed trachea (donor 4) and a distal lung (donor 1) biopsies with hematoxyllin and eosin (H&E) staining (up) coupled to the maps of cell types identified by HybISS on top of nuclei (DAPI, white) in the same sections (down). Spots represent detected transcripts, colored according to the corresponding cell type of the cell they were assigned to. Colors as in Fig. 1B. Dashed lines indicate the approximate borders of histologic compartments. SMG submucosal gland, aw airway, alv alveolar region, bv blood vessel. Scale bar 200 μm. **E** Cell type neighborhood enrichment graph representing cell types as nodes, and edges indicating a positive neighborhood enrichment (>2) between cell types across the profiled sections. Suggested neighborhoods are shown as bubbles. Node colors as in Fig. 1B. Source data are provided as a Source Data file.

Complementing the HybISS datasets, we profiled sequential sections by SCRINSHOT[16], employing a pre-validated gene panel of 64 marker genes[20]. This panel overlapped with HybISS by 43 genes. The gene counts per cell were obtained as for HybISS, and after clustering major cell types were assigned according to marker gene positivity (Supplementary Data 4). We annotated 24 cell types by clustering and also defined an additional cell state (AT0) based on previous publications reporting the co-expression of airway and alveolar epithelial markers[9] (Supplementary Fig. 4). Two of the cell clusters (immune nan and epithelial nan) could not be further subdivided due to the detection of only general and non-discriminative markers. Seven cell types contained multiple clusters or subclusters that were annotated as subtypes or states (Supplementary Fig. 5). We inspected and confirmed the signals and cell types on the tissue to ensure their expected location. We correlated spatial distribution of cell types in HybISS and SCRINSHOT by plotting them on serial sections and comparing their proportions in the same tissue compartments of serial sections (Supplementary Fig. 6). In addition, we analyzed sequential sections of two tissue blocks using the Visium protocol, followed by the Stereoscope method[23] to deconvolve the cell type composition of each spot using the finest annotation from scRNA-seq dataset from Madissoon et al[1] as a reference. Visual assessment confirmed that the location of most assigned cell types within the tissue was consistent between all three methods (Supplementary Fig. 7, A-B). Overall cell type proportion distributions across manually defined tissue compartments on serial sections correlated significantly between the three SRT methods (Supplementary Fig. 7C), confirming the specificity of each of the technologies. The combinatorial approach with three methodologies hand-in-hand overcomes limitations of individual spatial mapping protocols, such as cellular resolution or variable detection sensitivity.

## Specific cell types and states in distinct tissue compartments

In order to characterize cell type distribution, we defined cell type compositions across tissue locations by calculating the relative frequency of each cell type within each profiled region (Fig. 1C). We treated distal regions as a single location due to their similarity in cellular composition. Several cell types exhibited a regional preference, for example AT1 and AT2 epithelial cells were mostly present in distal lung, whereas B plasma and SMG cells mainly occupied tracheal regions, as expected (Fig. 1C, D). To further dissect the relative spatial distributions of cell types and describe consistent cellular colocalizations across the mature lung, we performed neighborhood proximity enrichment analysis of all datasets. This revealed multiple cellular colocalizations, including most cell types, except tuft and lymphatic endothelial cells. The colocalizations were combined into larger neighborhoods, which were characterized by defined histological features (Fig. 1D, E). In addition to the SMG and airway epithelial neighborhoods, we also revealed a group of cell types in proximity to AT1, AT2 and aerocytes, which included stromal, endothelial, monocytes and NK cells (Fig. 1E). This neighborhood was therefore labeled alveolar parenchyma. A distinct neighborhood composed of adventitial fibroblasts, venous and immune cells, was revealed in both peri-bronchial and peri-SMG locations (Fig. 1D, E). In summary, our combined data provide an overview of 35 cell types and their occurrences in the topographic map of healthy adult lung, defining distinct cellular neighborhoods based on cell type proximity in the entire tissue.

To complement predicted neighborhoods, we addressed common classification of tissue compartments using histologic landmarks and cellular morphology, similarly to a recently published approach[24]. We related cellular morphologies in hematoxylin-eosin (H&E) staining with cell-type annotations and gene expression on the same section. We defined peri-epithelial, such as (i) the SMG, (ii) airways, and (iii) alveolar compartment, and non-epithelial, such as (iv) veins, (v) arteries, and (vi) cartilage (Supplementary Fig. 8A-C, Fig. 2A, Methods). These histological subdivisions were largely in agreement with the

predicted neighborhoods (Figs. 1E, 2A–C) and covered most of the tissue area (Supplementary Fig. 8A-C). We mapped the 35 HybISS-based cell types and SCRINSHOT-based subtypes in relation to histologically defined tissue compartments, assessing compartment-specific gene expression in cell types.

First, we focused on the submucosal gland structure, which includes a duct protruding from the airway lumen branching into the tubules and acini composed of mucous and serous cells[25]. SMG mucous and serous cells expressing their corresponding markers (Supplementary Data 3 and 5, Supplementary Fig. 2D, Supplementary Fig. 4G) were either intermingled with each other or were found in continuous patches of either mucous or serous cells. In the ducts, we detected cells expressing airway secretory cell markers (*LCN2, ALDH1A3, SCGB3A1*) together with serous or rarely mucous markers (Supplementary Fig. 4G) and therefore called these cells SMG intermediate. These cells corresponded to SMG duct cells from scRNA-seq in SCRINSHOT dataset (Supplementary Fig. 4G-H). In contrast, *SCGB3A2* was expressed in a subpopulation of serous cells, usually located in small tubules and not in the duct (Supplementary Fig. 5C, Supplementary Fig. 6C)[10,26]. The previously reported description of an SMG immune niche[1], as well as our neighborhood analysis (Fig. 1E) suggests specific cell type enrichment around the gland. The peri-SMG non-epithelial compartment in the trachea and proximal lung was presented by B-plasma cells and fibroblasts (Fig. 2A, B), confirmed in Visium dataset, with *JCHAIN* (marker of B plasma cells), as well as *PLA2G2A* (adventitial fibroblast) expression (Supplementary Fig. 9). Cell subtype analysis in the SCRINSHOT dataset indicated the presence of fibroblasts expressing *FBLN1*, myoepithelial cells, T, NK and other immune cells, which could not be more precisely annotated (nan) (Fig. 2B and D, Supplementary Fig. 6D). Interestingly, endothelial cells around the tracheal SMG predominantly expressed all endothelial markers (*SPARCL1, IGFBP7* and *CLDN5*), potentially corresponding to pulmonary vasculature (venous/arterial/capillary), whereas in the lobes we found either *SPARCL1*, or *IGFBP7* or *CLDN5* positive cells potentially corresponding to systemic arterial/venous, lymphatic or capillary cells, respectively (Fig. 2B–D, Supplementary Fig. 5B-D, Supplementary Fig. 10). In conclusion, we identified unexpected heterogeneity of the epithelial cell populations in the SMG, and specific B plasma and adventitial fibroblast cells within its compartment.

As expected from the neighborhood analysis, the peri-airway compartment contained very similar cell type combinations as the peri-SMG one (Fig. 2B–D). However, the peri-bronchial compartments varied in different anatomic locations. For example, we only found histologic ganglia structures with *VIM/CD9* double-positive cells (annotated Schwann-like cells according to their morphology) in proximal lung (Fig. 2D, Data viewer for SCRINSHOT Atlas). Additionally, smooth muscle cells were most abundant in proximal bronchi, whereas the distinct populations of endothelial cells expressing either *SPARCL1* (which could indicate venous, aerocyte or arterial cells) or *CLDN5* (capillary or arterial) were only found in lobular bronchi. In distal lung, *APOE* expressing macrophages also appeared in peri-bronchial region (Fig. 2C, D). These data suggest high heterogeneity of gene expression along the proximo-distal axis of the airways.

The alveolar parenchyma was defined by the presence of AT1 and AT2 epithelial cells (Fig. 2A, Supplementary Fig. 8C) and was dominated by capillaries (including aerocytes), and general (non-adventitial) fibroblasts (Fig. 2A, C). In comparison to other compartments, the alveolar parenchyma had the highest proportion of *CLDN5* positive endothelial cells, most likely corresponding to alveolar capillaries, and *APOE* (alveolar) macrophages (Fig. 2C–D)[1,2]. Fibroblasts positive for *RGCC* (alveolar fibroblasts) were dominating in the distal lung (Fig. 2D). Fibroblasts in the distal lung also expressed higher *FN1* and *RGCC*, and lower *PLA2G2A* and *C3* levels, compared to the fibroblasts in the other regions (Supplementary Fig. 8D). These gene expression patterns suggest the existence of multiple fibroblast subtypes located in

**A**   **Cell type distribution by compartment**

**B**   **Peri-tracheal/SMG cell types**

**C**   **Peri-bronchial/alveolar cell types**

**D**   **Cell subtypes in peri-epithelial compartments**

**E**   **Peri-epithelial gene expression**

**Fig. 2 | Distribution of non-epithelial cell subtypes in histological tissue compartments. A** Heatmap of average (mean) cell type proportions (%) in manually annotated histological compartments using HybISS cell type maps. Proportion calculations include non-annotated cells, not presented in the heatmap. Cell type/subtype maps on top of nuclei (DAPI, white) of HybISS (left) and SCRINSHOT (right) datasets of donor 1 trachea (**B**) and distal lung (**C**), respresentative images from four biological replicates. Dotted outlines: peribronchial and airway (AW), submucosal gland (SMG), alveolar (Alv) and arterial (Art) compartments. Cell type abbreviations: pb peribronchial, adv adventitial, alv alveolar, pv perivascular, cap capillary, gen general, art arterial, ven venous, nan not annotated. Scale bar 200 μm. **D** Graph of mean cell type/subtype proportions in each of the peri-epithelial compartments. Floating bars indicate Min/Max values as bounds of box, with a line on mean

proportion per compartment, and indicated in percent (%), $n = 3$ in tracheal regions, $n = 4$ in proximal and distal lung regions. Values were compared in lung regions using Friedman two-sided test followed by Dunn's multiple comparisons test. Significant differences ($P < 0.05$) are indicated by asterisk (*). Adjusted P values are as follows: $P = 0.0370$ for fibroblasts (RGCC) between alveolar parenchyma and proximal bronchi; $P = 0.0370$ for smooth muscle cells between alveolar parenchyma and proximal bronchi; $P = 0.0370$ for endothelial (CLDN5) cells between alveolar parenchyma and proximal SMG. The cell subtypes that dominated in one anatomic location are highlighted with coloured boxes. **E** Schematic summary of the gene expression in the parenchyma of each compartment. Source data are provided as a Source Data file.

different peri-epithelial tissue compartments. Statistical analysis indicated that some cell types demonstrate significant variability between compartments. However, pairwise comparison test within each anatomical location did not give any significant results, when all cell types were compared simultaneously. We therefore can conclude that the majority of subtype gene expression is driven by anatomical location, rather than by the compartment within each location.

Among non-epithelial compartments, large vessels and cartilage contained endothelial cells expressing most endothelial markers, and *FBLN1* positive fibroblasts. The peri-arterial compartment was distinguished by the high proportion of smooth muscle cells (Supplementary Fig. 8E). The peri-venous compartment contained small proportions of all mesenchymal cell types. Chondrocytes were detected only in the Visium dataset (Supplementary Fig. 7A). The perichondrial regions were composed of *FBLN1* and *PLA2G2A* positive fibroblasts, and occasionally capillaries, pericytes, and mast cells (Fig. 2A, Supplementary Fig. 8E).

In order to define the differences between *SPARCL1* and *CLDN5* expression in the vascular system, we first estimated the general distribution of endothelial subtypes in SCRINSHOT and HybISS and defined that most of these cells are located in small vessels or capillaries and are of mixed origin (Supplementary Fig. 10A). Immunofluorescent staining confirmed that the vessels around the airways were surrounded by a thin layer of aSMA, indicating that they belong to venule and/or arteriole category, whereas CLDN5-positive vessels located in the alveolar region lacked aSMA staining, and belong to capillary network (Supplementary Fig. 10B). Therefore we concluded that the airways are surrounded by a network of small venules or arterioles, which are positive for SPARCL1 and low-positive for CLDN5. The cells expressing high levels of *CLDN5* are most likely general (non-aerocyte) alveolar capillary cell population.

Location-specific distributions of cell types and cell states with distinct gene expression patterns in different compartments define cell type niches and inform on potential cell-to-cell signaling domains. Our data reveal an enrichment of *APOE* macrophages and endothelial cells highly expressing *CLDN5* in alveoli. Peri-bronchial and peri-SMG regions on the other hand, were composed of *FBLN1* fibroblasts, and *JCHAIN* plasma cells, with *PLA2G2A* fibroblasts enriched around the gland and cartilage (Fig. 2, Supplementary Fig. 8E), which were also confirmed by our Visium dataset (Supplementary Fig. 9). The definition of regional gene expression variation in non-epithelial cell types, such as fibroblasts, immune and endothelial cells in the healthy lung provides a basis for the precise comparison of the same regions in the diseased states. This may distinguish the regional gene expression variations from the disease-associated ones.

## Multiple cell states with distinct topologies in the airway epithelium

We next focused on the topography of cell diversity in the airway epithelium. Among bronchial epithelial cells, a total of 11 cell types were identified, including basal, suprabasal, ciliated, deuterosomal, and neuroendocrine cells, as well as manually assigned ionocytes, tuft (brush), rare tuft-like and squamous-like cells (Fig. 1B, Supplementary Fig. 2B, Supplementary Table 2, Supplementary Data 3). The remaining bronchial epithelial cells were split into two groups. First, secretory cells expressing the *AGR2, SFTPB, BPIFB1, MUC5AC* markers and comprising 33% of total bronchial epithelial cells and second a smaller group comprising 12% of total bronchial epithelial cells, which were positive for the general epithelial marker gene *SLPI* (Supplementary Data 3). These cells were found spread along the airways, but also occasionally in SMGs and alveoli. However, they were negative for the characteristic epithelial cell type markers, such as mucins or secretoglobins and were designated not annotated 'nan' cells (Fig. 3A, Supplementary Data 3). They could represent less differentiated epithelial cells or unknown cell states.

To investigate cell type composition diversity along the airway proximal-distal axis, we further characterized the composition of the airway epithelium in tracheal, proximal and distal airway sections from individual donor samples by HybISS. In the trachea, the epithelium was dominated by suprabasal cells, whereas in the intralobar airways the epithelium was mainly composed of secretory and ciliated cells (Fig. 3A). Gene expression comparison across the regions confirmed this distribution, with basal and suprabasal (*KRT5, KRT15, S100A2*), and squamous (*SPRR3/1B*) genes expressed predominantly in the trachea. In addition, this analysis revealed further variability across the regions, including, for example, mesothelin (*MSLN*) expression in trachea, trefoil factor 3 (*TFF3*) and *SLPI* in the proximal lung, and surfactant protein B genes (*SFTPB*) in the distal lung (Fig. 3B). These variable genes could mainly be attributed to distinct secretory cell populations or regional variations in the secretory cell transcriptomes. Statistical analysis of all four donors confirmed a significant dominance of *AGR2*-positive populations in the trachea compared to other regions, and the higher abundance of *SFTPB*-positive populations in distal lung, compared to the trachea (Supplementary Fig. 11A). This analysis identifies consistent differences in epithelial composition between three anatomical regions along the proximo-distal axis of the airway tree. To further define the location of the major secretory cell subtypes, we quantified gene expression by SCRINSHOT, targeting characteristic cell type markers for goblet, club and terminal bronchiole epithelial cells (pre-TB or TASC or RAS)[10,13,27] in three different locations (Supplementary Data 4, 5). We found club cells in all three anatomical regions but localized goblet cells dominating in trachea and proximal lung and pre-TB cells only in distal lung[10,28] (Supplementary Fig. 11B). Similar to HybISS dataset, a group of epithelial cells could not be annotated by characteristic markers and was assigned epithelial nan (Supplementary Fig. 11B-C, Supplementary Data 5). Interestingly, subpopulations of the club cells co-expressed low levels of mucins. Moreover, in contrast to pre-TB cells, the club cells expressed genes encoding antimicrobial proteins (such as *LTF, LCN2*, and *CYP2F1*, Supplementary Fig. 5C), suggesting a specialized role in epithelial immunity. Both distal club and pre-TB cells were located in small clusters along distal bronchi and respiratory bronchioles (Fig. 3C). We also detected terminal respiratory bronchiolar (TRB) secretory and alveolar type 0 (AT0) cells in peri-bronchial and alveolar regions respectively (Fig. 3C). AT0 cells were defined by co-expression of the alveolar type II cell marker *NAPSA*, and low but evident levels of either *SCGB3A2*[9], or *SCGB3A1* or *LCN2*, suggesting additional heterogeneity in this cell type (Supplementary Fig. 5C, Fig. 3C). Overall our spatial analysis of cell type locations and distributions confirms the previously published proximo-distal variation in gene expression, and reveals that club cells are present in all anatomical regions at similar proportions, whereas other secretory cell populations are location-specific.

The analysis so far revealed large gene expression heterogeneity in the distal airway epithelium and a dominant abundance of suprabasal epithelial cells in the thicker tracheal epithelium (Fig. 3A–C). To investigate the suprabasal cell type further and define its topological relationships within the pseudostratified tracheal epithelium, we investigated gene expression in relation to the distance from the basal membrane to the lumen. In the HybISS dataset, basal and suprabasal cells were enriched close to the epithelial basement membrane, according to their nuclei locations. In contrast, secretory and ciliated cells were enriched in more apical positions, ciliated cells being closest to the airway lumen (Fig. 3D, Supplementary Fig. 11D). Quantification of the mRNA signals along the distance from the basal membrane to the lumen defined basally-enriched (*KRT15, IFITM1* and *IFITM2*), and apically-enriched mRNAs (*BPIFB1, CAPS*), as well as an intermediately located gene expression program (*SERPINB3* and *HSPB1*)[29] (Fig. 3E). In addition, *S100A8* and *SPRR1B, SPRR3* were variably expressed in different donors in scattered distances from the basal membrane (Supplementary Fig. 11E-F), which could be related to epithelial condition[30].

## Variations along proximo-distal axis

**A** Cell type frequencies in bronchial epithelium

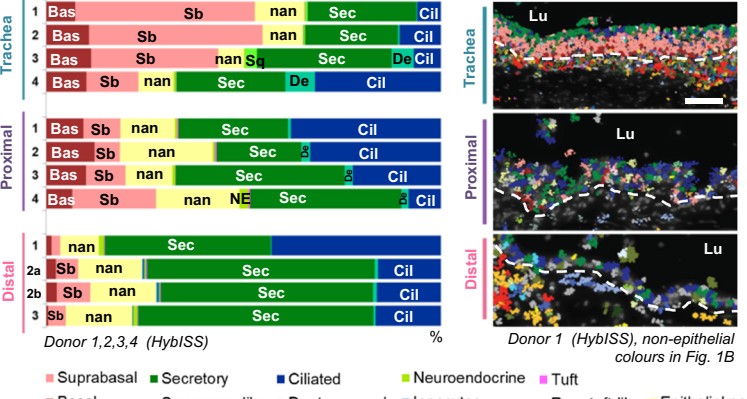

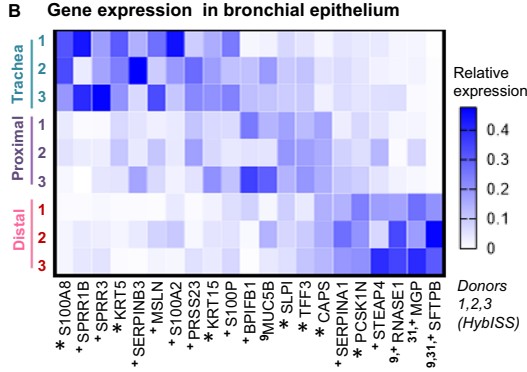

Donor 1,2,3,4 (HybISS)

Donor 1 (HybISS), non-epithelial colours in Fig. 1B

Legend:
- Suprabasal
- Basal
- Secretory
- Squamous-like
- Ciliated
- Deuterosomal
- Neuroendocrine
- Ionocytes
- Tuft
- Rare tuft-like
- Epithelial nan

**B** Gene expression in bronchial epithelium

*S100A8 +SPRR1B +SPRR3 *KRT5 +SERPINB3 +MSLN *S100A2 +PRSS23 *KRT15 *S100P +BPIFB1 9MUC5B *SLPI *TFF3 *CAPS +SERPINA1 *PCSK1N +STEAP4 9,+RNASE1 31,+MGP 9,31,+SFTPB

Relative expression: 0.4 0.3 0.2 0.1 0

Donors 1,2,3 (HybISS)

**C** Distribution of secretory cell subtypes

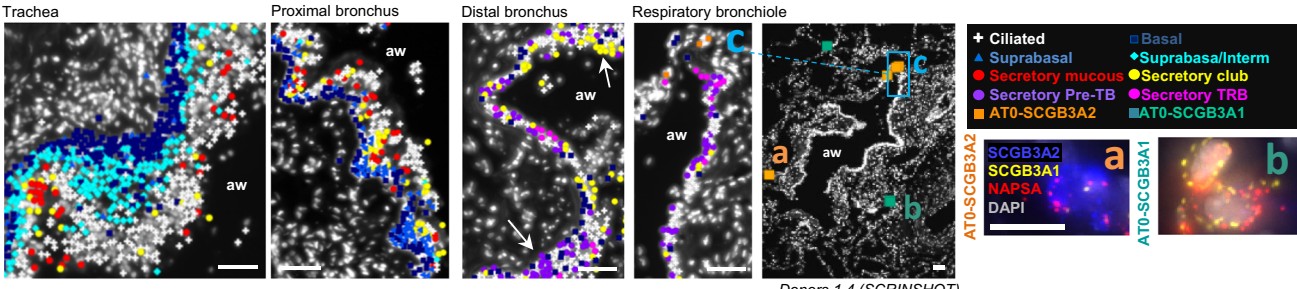

Donors 1,4 (SCRINSHOT)

Legend:
- + Ciliated
- ▲ Suprabasal
- ● Secretory mucous
- ● Secretory Pre-TB
- ■ AT0-SCGB3A2
- ■ Basal
- ◆ Suprabasal/Interm
- ● Secretory club
- ● Secretory TRB
- ■ AT0-SCGB3A1

SCGB3A2 / SCGB3A1 / NAPSA / DAPI

## Variations along apical-basal axis

**D** Cell type distribution

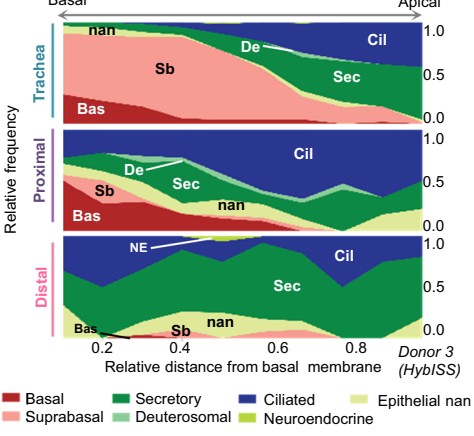

Donor 3 (HybISS)

Legend:
- Basal
- Suprabasal
- Secretory
- Deuterosomal
- Ciliated
- Neuroendocrine
- Epithelial nan

**E** Gene expression distribution in epithelial cells in trachea

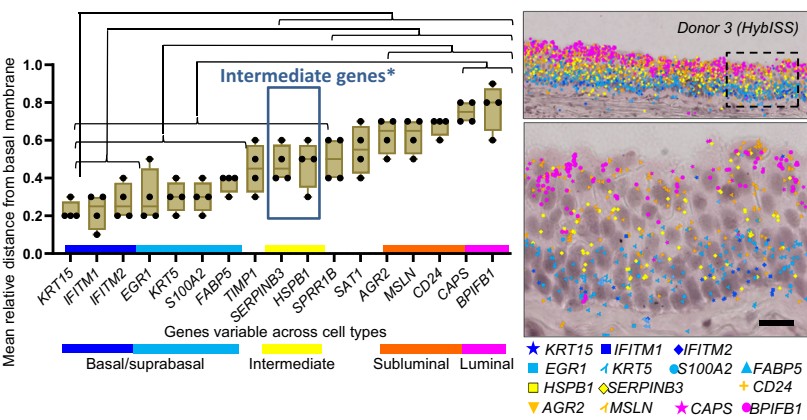

Intermediate genes*

Genes variable across cell types: Basal/suprabasal — Intermediate — Subluminal — Luminal

KRT15 IFITM1 IFITM2 EGR1 KRT5 S100A2 FABP5 TIMP1 SERPINB3 HSPB1 SPRR1B SAT1 AGR2 MSLN CD24 CAPS BPIFB1

Donor 3 (HybISS)

Legend:
- ★ KRT15
- ■ IFITM1
- ◆ IFITM2
- ■ EGR1
- ◢ KRT5
- ● S100A2
- ▲ FABP5
- ■ HSPB1
- ◇ SERPINB3
- + CD24
- ▼ AGR2
- ◢ MSLN
- ★ CAPS
- ● BPIFB1

SCRINSHOT analysis of consecutive tracheal sections targeting *S100A9* and *KRT13* detected their expression in the intermediate and apical layers, and *SERPINB3* in the intermediate layer (Supplementary Fig. 11G). Only a subset of *S100A9* cells expressed *KRT13* (Supplementary Fig. 11G). This supports that the *KRT13* positive cells might correspond to hillock cells, a distinct tracheal cell state, in line with recent observations[31]. Our spatial analysis reveals multiple cell states located in layers of the pseudostratified tracheal epithelium. Overall, there is a strong correspondence to three layers along apical-basal axis of the airway epithelium, with characteristic *HSPB1* and *SERPINB3* expression in the intermediate layer, distinct from previously characterized hillock and squamous cell states[3,7,31], which in this study appear donor- and layer-variable (Supplementary Fig. 11H).

## Rare cell type mapping reveals region-specific neuroendocrine cells

The variability of gene expression patterns in the apical epithelial cells along the proximo-distal axis could partly be explained by differential exposure to external factors. Certain environmental factors are sensed by specialized rare cells located in the airways. Ionocytes and tuft cells have been identified in the nasal epithelium and distal airways, whereas neuroendocrine cells were predominantly located in trachea and intermediate airways[3]. Our HybISS-based analysis allowed mapping most of these cell types. Yet, the expression levels of their markers were low, making it difficult to extract safe conclusions regarding their differential distribution. To explore the potential variation and location of rare cells, we developed a marker panel targeting pulmonary ionocytes, tuft-like cells and neuroendocrine (NE) cells, based on the

**Fig. 3 | Distribution of cells and gene expression in bronchial epithelium along proximo-distal and apical-basal axes. A** Cell type frequencies in the airway epithelium according to HybISS. Left: stacked bar plot representing relative frequencies of the airway epithelial cell types across regions from different donors, using samples with representative numbers of airway cells. Right: cell type maps in the indicated regions from donor 1, respresentative images from four biological replicates. Spots represent detected transcripts, colored according to the corresponding cell type of the cell they were assigned to. Colors as in Fig. 1B. Nuclei: gray. Lu lumen, dashed line: approximate location of basal membrane. Scale bar: 50 μm. **B** Heatmap of the relative mean gene expression in airway epithelium of variable epithelial markers across regions (colors) in three analysed donors (numbers). The expression is normalized by gene, dividing by sum of values in each row. Superscript numbers: references of previous studies, reporting variable expression of the corresponding marker along the proximal-distal axis. Asterisk*: statistically significant expression differences of the corresponding marker between regions ($P < 0.05$, repeated measures ANOVA with Geisser-Greenhouse correction, followed by Tukey's multiple comparisons test, all 161 detected genes tested). Plus +: having highest mean change. **C** Maps of epithelial cell types detected by SCRIN-SHOT in the indicated regions of the airways, respresentative images from four

biological replicates. Arrows: cell clusters. Inserts in respiratory bronchiole map: (a-b) SCRINSHOT images of representative AT0 cells with either *SCGB3A2* (a, orange squares) or *SCGB3A1* (b, jade squares) dominating expression. (c) Zoomed area of respiratory bronchiole. Nuclei: gray. Scale bar in maps: 50 μm. Scale bar in SCRIN-SHOT images 10 μm. aw airway. **D** Area plots representing the relative apical-basal cell type distribution across regions according to HybISS (data from one representative donor 3). X axis: relative distance of cells from the basal membrane. Y axis: relative frequency of cell types. **E** Mean gene expression of significantly layer-variable markers along the apical-basal axis of the tracheal epithelium ($n = 4$). Individual data points represent biological replicates. Box bounds indicate standard deviation, line indicates mean, and whiskers indicate minimum and maximum values. Significant differences ($P < 0.05$) between gene dot distances are calculated using nested one-way ANOVA followed by Tukey's multiple comparisons test, and indicated with lines and brackets. Exact *P* values are indicated in Source Data. Blue box* indicates the genes that are located differently from both basal and luminal values. To the right, image of HybISS-detected transcripts in tracheal epithelium, where each transcript is shown as a characteristic coloured shape. The image is representative of four biological replicates. Squared area in the image is shown magnified below. Scale bar 20 μm. Source data are provided as a Source Data file.

previously integrated human lung cell atlas from scRNA-seq[2], as well as specific airway epithelial cell types[3] and embryonic single-cell atlas studies[32,33]. Since neuroendocrine cells of adult lung are diverse, a precise selection of markers was performed to uncover potential heterogeneity in neuroendocrine cell phenotypes, targeting the four most abundant adult NE genes, as well as two genes marking a NE population discovered predominantly in the developing embryonic lung[8,32]. We used these markers in SCRINSHOT and located rare epithelial cell types manually by positivity for the expected markers. We assessed gene expression in 180 rare epithelial cells from four donors, and clustered these cells, identifying at least four groups of neuroendocrine cells: (1) NE-GRP, expressing *GRP* and low levels of *ASCL1*, (2) NE-ASCL1 expressing *ASCL1* and low levels of *GRP*, (3) NE-GHRL positive for *GHRL* and *CFC1*, and (4) NE-PCSK1N expressing variable levels of *PCSK1N, GRP* and *ASCL1* (Fig. 4A, B, Supplementary Fig. 12A-B). All NE groups sparsely expressed variable levels of *CHGB* and were represented in each donor. GHRL and GRP differential location was confirmed by immunofluorescence (Fig. 4C). *ASCL3* expression defined ionocytes, and cells expressing variable levels of *POU2F3*, *RGS13* and *CRYM* were annotated as tuft cells, the latter including a rare tuft-like cell population expressing previously published markers *NREP* and *HES6*[3] (Fig. 4A, B, Supplementary Fig. 12A-B). We visually analyzed samples from three regions of four donors and selected samples with large parts of the airway (covering a continuous airway length of at least 2 mm per section) for further quantification. In order to create a uniform regional annotation of their positions disregarding the variable epithelial thickness, we quantified rare cell types per length of basal membrane from the selected samples from at least two donors per region (Supplementary Fig. 12C) and then assessed in proportion to the remaining airway epithelial cells (Fig. 4D). Ionocytes were observed in all anatomical locations, preferentially in trachea and proximal bronchi (Fig. 4A, D). Tuft and rare tuft-like cells were mostly located in proximal bronchi, but were observed in other locations along the airway, occasionally in close proximity to other rare cells, but also solitary (Fig. 4A, D). The three neuroendocrine cell identities were observed across locations, but interestingly, GHRL-positive NE cells only appeared in distal bronchioles of three donors and were not observed in trachea or proximal lung (Fig. 4A, D, Supplementary Fig. 12D). GHRL-positive NE cells have previously been detected in embryonic and pediatric datasets and these cells were hypothesized to gradually disappear in adulthood[33,34]. Our results indicate that targeted spatially-resolved methods allow the detection of low abundant or very rare cell populations with high efficiency, enabling the evaluation of the roles of these cells in lung function.

## Spatial analysis of early-stage COPD samples demonstrates AT0 cell alterations and new cellular neighborhoods

We further explored the utility of our topographic atlas as a reference to detect deviations in cellular proportions, gene expression and neighborhoods in diseased lung tissue. We focused on chronic obstructive pulmonary disease (COPD), a common lung disease affecting airways and alveoli, using samples from 3 patients with COPD GOLD stage II obtained from the most distal lung locations (corresponding to region 3c in Fig. 1A). These samples were derived from the tumor-free regions from lung cancer surgeries. Two healthy atlas samples together with one histologically normal tumor-free lung sample of a chronic lung disease (CLD)-free cancer patient were processed side-by-side for comparison. Samples contained variable airway sizes (large, medium, small bronchioles and respiratory bronchioles). We applied a modified SCRINSHOT panel to test the expression of the 41 most selective genes in order to define major cell types (Supplementary Fig. 13A).

We annotated 84,631 high-quality cells and defined 20 major cell types according to their markers, regardless of their surroundings (Supplementary Fig. 13B). The major cell classes were equivalently represented in all analyzed samples (Supplementary Fig. 13C), however the proportion of AT1 cells was decreased and the proportion of T lymphocytes increased in COPD samples (Fig. 5A). Previous extensive scRNA-seq studies of COPD samples reported a shift in the expression of epithelial secretory cell gene programs, where proximal airway gene expression levels gradually increased in distal epithelial cells of COPD airways[35] leading to a decrease in the proportion of pre-TB (TASC) secretory cells[10]. Recent publications reported an increase in bronchial secretory cell type marker expression in the AT2 cells from COPD patients[27], and an increase in AT0 cell state[9]. We therefore subclustered both bronchial secretory and AT2 cells and defined a population of cells co-expressing AT2 (*NAPSA, SFTPC*) and airway (*SCGB3A1, SCGB3A2, LTF, LCN2*) markers, which we annotated as AT0 cells (Supplementary Fig. 13B, D). We found that the proportion of these AT0 cells was significantly increased in all COPD samples (Fig. 5A, B). Histological compartment analysis indicated that these AT0 cells were mostly in the alveolar regions in proximity to the large and small airways, and in respiratory bronchioles, but the increase primarily affected alveolar regions (Supplementary Fig. 14A-E). This in situ increase in AT0 state is in line with the scRNA-seq analysis arguing for a general upregulation of the proximal secretory cell type program in epithelial cells, not only in the airways, but also in the alveoli[27,35].

We extended our analysis to find potential COPD-specific cellular niches. First, we compared healthy and diseased peri-bronchial and

**Fig. 4 | Rare cell types and their distribution in the airways. A** Maps of rare cell types detected with SCRINSHOT from three anatomical regions (donor 4), respresentative images from three biological replicates. Scale bar 200 μm. Lu lumen. Raw SCRINSHOT signal images of cells labelled with letters (a-f) are shown in Supplementary Fig. 12B. **B** Heatmap of gene expression demonstrating unique and overlapping marker genes within the detected rare cell types from four donors. At least two out of four donors demonstrated each cell type in each anatomical region, except NE-GHRL population. Number of cells quantified (180): ionocytes−41, tuft-like:−27 (including tuft−24 and rare tuft-like−3), NE-GHRL−9, NE-PCSK1N−32, NE-ASCL1−40, NE-GRP−31. **C** Immunofluorescent staining for GHRL (green) and GRP (magenta) of two subtypes of neuroendocrine cells, as well as epithelial membrane marker CDH1 (white) on top of nuclei (DAPI, blue), distal lung of donor 4, which had largest number of neuroendocrine cells from three stained samples. Areas (a) and (b) are crops from a larger image in Supplementary Fig 12D. Scale bar: 10 μm. **D** Bar plot of the number of the detected cells per total airway epithelial cell number (shown in %) from four donors, error bars: standard deviation. Individual data points represent biological replicates (only donors with airway larger than 2 mm of basal membrane length). Arrow: NE-GHRLpos cells appearing only in distal lung. Source data are provided as a Source Data file.

alveolar non-epithelial cells and found the increase in T lymphocytes in both COPD compartments, and NK cells in peri-bronchial, with a decrease in *CLDN5* endothelial and *RGCC* fibroblasts in alveolar compartments (Supplementary Fig. 15A). Following this, we performed neighborhood analysis (Methods)[36], and clustered the COPD-cellular neighborhoods together with the ones from the healthy atlas, which contained the coordinates of 218,496 cells, 164,719 of which were grouped into 30 annotated cell types, from three anatomic locations (3-4 donor samples per location). The integrated data separated into twelve neighborhood clusters with two of them corresponding to the SMG, and ten of them matching the distal lung regions (Fig. 5C). Three of these neighborhoods were composed predominantly of cells

**A** Cell type proportions

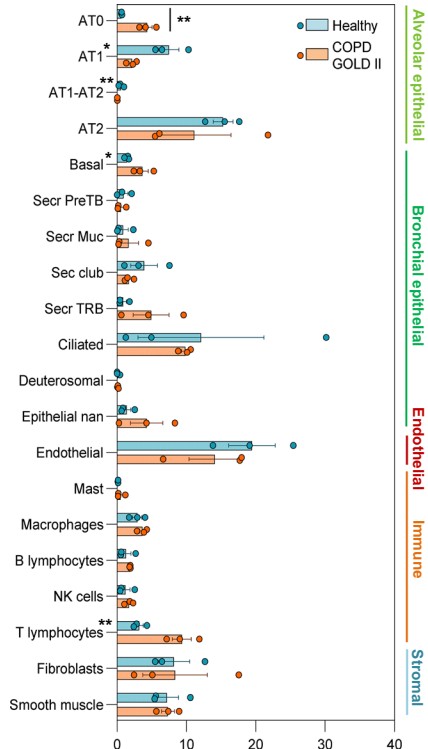

**B** Alveolar epithelial cell distribution

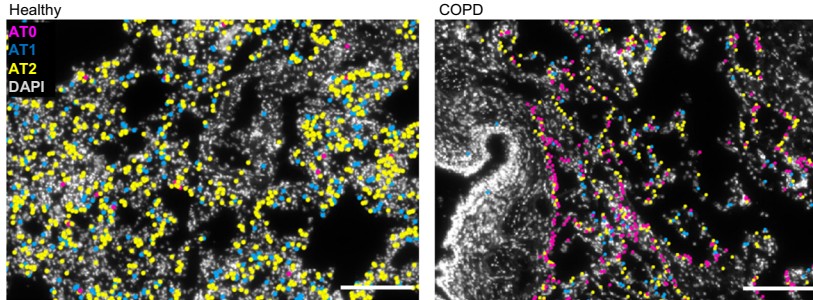

**C** Healthy and COPD cellular neighborhoods

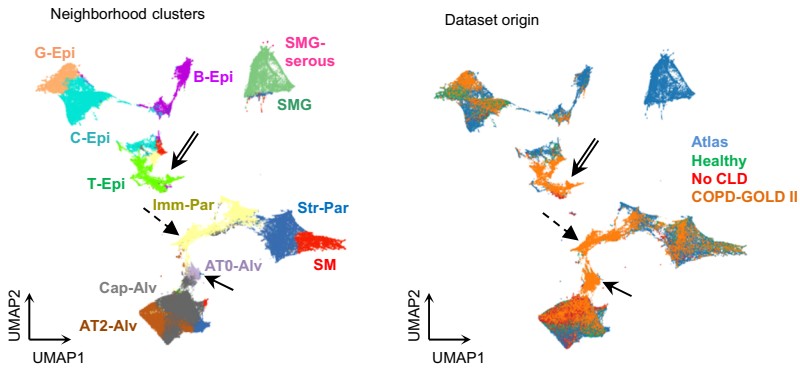

**D** Cell types within neighborhoods

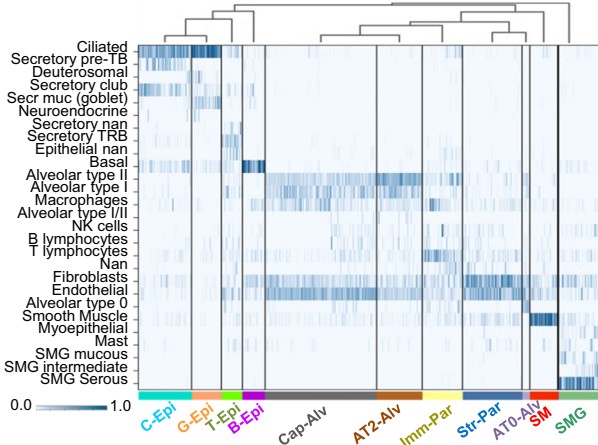

**E** Neighborhood spatial distribution

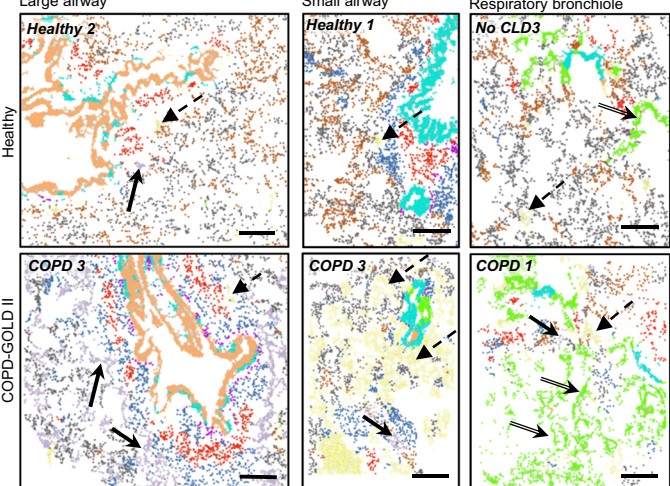

**Fig. 5 | Cell type and neighborhood changes in COPD. A** Box plot of the cell type numbers in healthy and COPD (mean ± standard error, with individual values). Significant differences are highlighted with asterisk (**, adjusted $P = 0.0087$), according to unpaired parametric two-sided multiple t-test (20) of logit-transformed data with Holm-Sidak correction. Individual data points demonstrate samples from different donors and patients, $n = 3$. The direct cell type comparison without the correction also revealed significant differences between healthy and COPD samples in the following cell types: AT0 ($P = 0.0004$), AT1 ($P = 0.0127$), AT1-AT2 ($P = 0.0090$), basal cells ($P = 0.0295$), T lymphocytes ($P = 0.0088$), labeled by asterisks at the cell type name. **B** Representative (from three biological replicates) maps of alveolar epithelial cell types detected in healthy and COPD samples, nuclei: gray. Scale bar 200 µm. **C** UMAP plots of analyzed cells labeled according to their cellular neighborhoods (left) and their corresponding condition (right). Neighborhood annotations: Cap-Alv capillary-enriched alveoli, AT2-Alv AT2-enriched alveoli, AT0-Alv AT0-enriched alveoli, C-Epi club-enriched epithelium, G-Epi goblet-

enriched epithelium, T-Epi secretory TRB-enriched epithelium, B-Epi basal cell-enriched epithelium, Imm-Par immune cell-enriched parenchyma, Str-Par stromal cell-enriched parenchyma, SM smooth muscle, SMG submucosal gland, SMG-serous serous cell-enriched submucosal gland, TRB terminal respiratory bronchiole. Arrows indicate the clusters that are predominantly composed of COPD-derived cells. **D** Heatmap exploring the cell type composition of each neighborhood, with cells (vertical lines) grouped by their assigned neighborhood cluster (x-axis) represented in Fig. 5C (left). Bar color represents the ratio of neighborhood enrichment with each cell type for each cell. **E** Maps of cell type neighborhoods in healthy and COPD samples in alveolar region with large and small airway and respiratory bronchioles. Color code as in C (left). Scale bar 200 µm. Simple arrows: AT0-enriched alveolar neighborhood, dashed arrows – immune-enriched parenchyma, double-line arrows – secretory TRB-enriched epithelium. Source data are provided as a Source Data file.

deriving from COPD samples of all three patients (Supplementary Fig. 15B, Fig. 5C, arrows, Supplementary Fig. 15C, arrows). The first COPD-cell neighborhood (termed T-Epi) located in terminal bronchioles and alveoli, contained cells from all disease-samples expressing TRB and AT0 markers, an unannotated secretory epithelial cell type, AT1 cells and endothelial cells (Fig. 5D). This neighborhood was particularly increased in proportion in one of the patients (Supplementary Fig. 15B, COPD II-1). The second COPD-specific cellular neighborhood (AT0-Alv) was composed of cells expressing AT0, AT1 and AT2 cell markers, fibroblasts and endothelial cells (Fig. 5D). This neighborhood mainly contained cells from another COPD patient (Supplementary Fig. 15B, COPD II-3). Finally, the third COPD-cell neighborhood (Imm-Par) was composed of T lymphocytes and other immune cells, fibroblasts, endothelial and AT2 epithelial cells, and was consistently increased in all three patients analyzed. (Fig. 5D, E, Supplementary Fig. 15B-D). This is in accordance with the known inflammatory nature of the COPD. Moreover, we identified two neighborhoods (AT2-Alv) and (Cap-Alv), composed of alveolar epithelial cells, endothelial cells, macrophages and fibroblasts, which were decreased in all three COPD patients compared to the healthy lung tissue samples (Fig. 5C-E, Supplementary Fig. 15B-D). This is consistent with the onset of alveolar simplification, which is an important component of the COPD pathology. This spatial analysis based on the topography of the healthy lung describes deviations in the cellular locations and neighborhood composition in the diseased lungs. We detected a reduction in the alveolar epithelial neighborhoods (AT2-Alv and Cap-Alv). Instead, AT0 cells in COPD patients were increased and contributed to different COPD-specific neighborhoods, AT0-Alv, T-Epi and Imm-Par (Fig. 5D, E). The neighborhood analysis reveals a consistent shift in the balance of the distal airway and alveolar cell phenotypes. We conclude that the usage of the spatially resolved healthy lung cell atlas as a reference aids the detection of cellular composition and cellular environments of tissue samples derived from diseased lungs.

## Discussion

We generated a reference topographic atlas of three representative regions in the adult human lung from four healthy donors, which included representatives of different age, sex and smoking status, yet only of one ethnicity. The histologically pre-assessed samples were analysed with ST, combining Visium unbiased transcriptomics with expression maps at cellular resolution for 162 genes with HybISS and 64 genes with SCRINSHOT. The targeted methods mapped 478,894 cells defining the location of 35 major cell types and discovered additional region-related cell variability. This variability includes new epithelial cell states, such as the *S100A8/9* cell state in the trachea, which included a *KRT13*-positive cell subset, and *SERPINB3* and *HSPB1* positive populations in the intermediate layer of the epithelium. We detected *CLDN5* expression in endothelial cells dominating in alveolar capillaries, whereas *SPARCL1* was expressed in larger vessels. We revealed a *GHRL*-expressing neuroendocrine cell group selectively in the distal airways, in contrast to their proximal location in embryonic lung development[32]. The utilization of complementary methods with different sensitivity, multiplexity and cellular resolution was also suitable to localize previously known but underrepresented cell types, including Schwann-like cells. However, a few of the expected cell types and subtypes could not be identified either due to the sparse signal from some probes, or due to the absence of particular cell types in the collected sections. Sparsity of signal, either from low gene expression, inefficient hybridization in thick tissue regions, or unsupervised probe selection, resulted in the presence of not annotated cells (nan). In addition, nuclei expansion-based segmentation resulted in the appearance of doublets, that could not be assigned to a specific cell type and were also labeled nan. Our analysis was also limited by the lack of certain cell types in the scRNA-seq datasets (such as neutrophils

and other granulocytes, and adipocytes). We detected these cells in the histological sections, but cannot confidently annotate them by gene expression. In addition, a number of genes in both SCRINSHOT and HybISS showed weaker signal than expected. Yet there were strong marker genes in the designed panels that enabled the detection of most cell types, providing correlation between all three spatial methods.

Our further analysis of cell neighborhoods relates gene expression levels with cell proximities and tissue histology and thereby facilitates future work to reveal homeostatic cell communication patterns by mining the extensive scRNA-seq datasets from healthy lung tissues[2]. A limitation of our study is the lack of statistical analysis relating to the variability in age, sex and smoking status of the donors. Despite the limited donor number and including individuals with different physiological status, we observed consistent cell types, states and regional specific neighborhoods in all of them. A major aim of the Human Cell Atlas projects is to generate reference maps and databases that also enable the study of diseased cell states[36,37]. We generated an additional SCRINSHOT dataset including two distal lung regions from the atlas, three from COPD stage II patients and a corresponding healthy region from another patient to explore the utility of our spatial atlas in identifying alterations in cell composition in disease. In this integration we used a general cell type panel, without aiming to target COPD-specific gene expression changes but to define a diseased phenotype purely by cellular neighborhoods and regional cell composition. Extensive previous work with samples from large cohorts indicated that COPD pathology is linked with multiple perturbed cell types[10,27]. Our relatively small experiment identified gene expression alterations including the increased proportion of AT0 and a relative reduction in AT1 cells in the diseased lungs, in line with previous knowledge of the COPD cell states and histopathology[9,35]. The active role of AT0 cell state in other diseases[9] may be relevant to the repair processes in the early COPD stages, however their proliferative function is shown be impaired in late COPD stages[38]. We further constructed the proximity-based COPD cellular neighborhoods and compared them with the atlas revealing aberrant, COPD-characteristic cell niches involving fibroblasts, macrophages, epithelial, endothelial and immune cells. Our identification of the cell types involved in the aberrant disease-specific niches provides a basis for hypothesis building and further exploration of available RNA-seq data from the corresponding cell types, as well as Visium data from COPD patients[10,27,39,40]. The extent and open availability of our atlas for exploration and data mining together with the sensitivity and versatility of targeted methods suggest their further application in the analysis of cell type composition and gene expression changes in different regions of the healthy and diseased lungs.

## Methods
### Sample collection and screening

**Donor information.** For the healthy adult lung atlas, four deceased organ donors of various age, gender and smoking status, without any previously documented lung-related conditions, were used. Informed consent from the families and approval from NRES Committee of East of England, Cambridge South, was obtained (15/EE/0152). Donors were anonymized and numbered randomly. Donor information can be found in Supplementary Data 1. For the diseased sample analysis, peri-tumor tissues were collected with written informed consent from all patients and with ethics approval from the ethics committee of the Ludwig Maximilian University of Munich (#330-10, #19-629 and #19-630). Patients admitted to the hospital due to lung cancer were either diagnosed with COPD or were COPD-free. Lobectomy of the lobe with the tumor was performed, and tissues were collected from peri-tumor (tumor-free) regions. Patients were anonymized and numbered randomly. Patient information can be found in Supplementary Data 1.

**Sample collection, freezing, and histopathological assessment of donor samples.** Tissues for the healthy lung atlas were collected from the lungs of deceased organ donors. Lung tissue samples (approximately 0.5-1 cm³) were collected from the following locations: (1) ventral side of trachea in proximity to carina, (2) left bronchus 2-3 generation, (3a) bottom part of left upper lobe, 1-2 cm from pleura, (3b) top part of left upper lobe, 1-2 cm from pleura, (3c) bottom part of left lower lobe, 1-2 cm from pleura. The samples were rinsed in PBS and stored until freezing in OCT within 2 hours from collection. The samples from COPD and non-COPD patients were collected from the following locations: (3c) left lower lobe, and (3d) right lower lobe. The samples were briefly rinsed in PBS and dried, then refrigerated until freezing in OCT within 18 hours from collection. Frozen blocks of tissue were sectioned using a cryostat at 10 μm thickness, fixed with 4% PFA and stained using hematoxylin and eosin (H&E). Imaged sections were evaluated by a histopathologist, samples with signs of severe inflammation were excluded from healthy cohort. A total of 55 tissue samples from six donors were sectioned and analyzed histologically. We excluded two donors as unhealthy due to observed pathological landmarks in alveolar regions. The remaining samples (from four donors described in Supplementary Data 1) were further screened using H&E staining for the presence of the airways, submucosal glands, alveoli and blood vessels, as well as the absence of freezing artifacts and other more subtle pathological conditions, such as inflammation or fibrosis. Within the patient cohort, six tissue samples from six patients (three COPD and three non-COPD) were histologically scanned. In order to create representative data from each of the three anatomic locations, we collected three locations from four donors (a total of 12 regions). For HybISS additional distal lung samples from a different lobe (for donors 2 and 3) were used. For SCRINSHOT we selected samples from all 12 regions that contained histological structures of interest, namely airways, alveoli and SMG. One of the tracheal samples (donor 3) was excluded from the analysis due to suboptimal mRNA quality of the sample. For Visium one representative sample from each anatomic region was processed. Two non-COPD samples were excluded due to poor quality and lack of airways. One remaining non-COPD sample together with two healthy donor samples were used for the experiment side-by-side with COPD samples. We processed morphologically suitable samples using SCRINSHOT and screened for RNA integrity with the presence of mRNA signals of well-characterized cell type marker genes. All samples are stored and available for further analysis.

**mRNA quality assessment.** For Visium/RRST eight 10 μm sections of each tissue were collected for the total RNA integrity (RIN) values. Samples with RIN values above 5 were processed further. For targeted spatial analysis one section per morphologically assessed tissue sample was processed with SCRINSHOT as described previously[16] for main cell type marker genes. The cell type marker panel was used for the detection of major cell types, as described previously[20]. Samples or regions with low or sparse signal were considered to be of unsuitable quality. Samples with specific patterns of gene expression and strong (>10 dots per cell) SCRINSHOT signal were considered suitable for further analysis. Selected samples are summarized in Supplementary Table 1.

## Gene panel selection
The cell type probe panels were designed using Spapros, as described previously[20]. For the SCRINSHOT panel, we utilized a precursor version of the method, which exhibited distinct characteristics compared to the current versions v0.1.0-v0.1.4. Specifically, during the training for binary cell type classification, individual decision trees were computed instead of generating multiple trees and subsequently choosing the most optimal one. Moreover, secondary trees aimed at enhancing classification performance for finely annotated cell states, which are

challenging to discern, were not included. The implemented version combined gene set selections for the lung regions: proximal (airway) and distal (alveoli, parenchyma) lung. This process involved choosing common genes for shared cell types and unique genes for region-specific cell types. Fifty genes for each region were chosen from the scRNA-seq lung data from previous publication[1]. Selections were based on log-normalised data post scran normalisation[41]. We omitted Donor A47 due to missing location annotations. An internal marker list was provided as input for Spapros, ensuring that genes from marker list groups that were not well captured with the initial selection were automatically included. For the SCRINSHOT panel gene selection, 26 clusters (23 cell types and 3 subtypes) were targeted. To filter out genes that might be below the detection threshold, an expression penalty was applied. This penalty employed a smoothed rectangular function to penalise genes with 0.99 expression quantiles below 2 and above 6 (set parameters on cpm log-normalised data), which translates to 0.75 and 4.3 respectively when adjusted to scran normalisation. Conversely, for HybISS selections, Spapros v0.1.0 was used on the same dataset but with more detailed cell type annotations covering 52 cell types (36) and subtypes (16). No region-specific selection or expression restrictions were applied. As before, the internal lung marker list was supplied for selection.

## In situ sequencing (HybISS)
**HybISS mRNA detection.** Cell type markers were selected as previously reported[19], replacing some of the markers by alternative markers when the design of specific padlock probes was not possible. The final panel of genes profiled can be found at Supplementary Data 2. The CARTANA High-Sensitivity library preparation kit was employed, following the manufacturer's instructions, with customized backbones as described in Supplementary Data 2. In the experimental process, tissue sections were fixed and then subjected to an overnight incubation with the probe mix in a hybridization buffer. Subsequently, stringent washing was performed, followed by incubation with the ligation mix. After further washes, RCA (Rolling Circle Amplification) was conducted overnight. For detection, labeling was performed according to the procedure described in the protocols.io website (https://doi.org/10.17504/protocols.io.xy4fpyw).

**HybISS imaging.** RCPs were detected using 4 different fluorophores (Cy3, Cy5, AF750 and AF488) across five imaging rounds. DAPI staining was imaged on each cycle to identify cell nuclei. All images were acquired using a Leica DMi8 epifluorescence microscope, which was equipped with various accessories. The microscope setup included an external LED light source called Lumencor® SPECTRA X light engine, an automatic multi-slide stage (LMT200-HS), a high-quality sCMOS camera named Leica DFC9000 GTC, and different objectives such as HC PL APO 10X/0.45, HC PL APO 20X/0.80, and HCX PL APO 40X/1.10 W CORR. For capturing multispectral images, the microscope was equipped with specialized filter cubes capable of separating 6 different dyes. Additionally, an external filter wheel (DFT51011) was used to enhance the imaging capabilities further. The image scanning process involved outlining Regions of Interest (ROIs) that could be saved for multi-cycle imaging, employing tiled imaging with a 10% overlap. To capture the depth of the tissue, Z-stack imaging was performed, covering 10 μm at 0.5 μm intervals.

**HybISS preprocessing and decoding.** The initial preprocessing of microscope images involved several steps. Z-stacks were subjected to maximum intensity projection, tiles were aligned between imaging cycles, and image stitching was performed. The code for this pre-processing can be accessed at https://github.com/Moldia/ISS_preprocessing. During imaging, the images and their accompanying metadata were exported. The images were then formatted into OME tiff files and stitched and aligned using ASHLAR[42]. To reduce

computational requirements, the stitched images were sliced into 6000 by 6000-pixel sections for decoding. Due to the heightened sensitivity of the High Sensitivity Cartana kit, we encountered challenges related to overlapping RCPs (optical crowding) in the 2D projected data, which was essential for decoding. To address this issue, we employed a content-aware image restoration (CARE) approach, which had previously been trained on pairs of raw-deconvolved RCP images of multiple tissues (https://github.com/Moldia/ISS_CARE)[43]. As a result, the RCPs in the CARE-processed images exhibited significantly enhanced sharpness, leading to a reduction in overlapping RCPs and improved decoding results. The transcript decoding process relied on the Python package called starfish (https://spacetxstarfish.readthedocs.io/en/latest/). The decoding code can be found at https://github.com/Moldia/ISS_decoding. In brief, the images were registered and underwent white top hat filtering. The channel intensities were then normalized across channels. Spots, representing transcripts, were located in a composite maximum intensity projected image of signal images from the same field-of-view and sequencing round. The decoding of spots was achieved using the PerRoundMaxChannel method. For each spot and each base, the highest intensity channel was determined and matched to a corresponding barcode in the codebook. Additionally, a quality metric was assigned to each spot in every cycle. This metric was defined as the called channel intensity divided by the sum of all other channels, with values ranging between 0.25 and 1.

**HybISS data analysis.** Cell segmentation was performed on each sample using the BIAS lite software (Single Cell Technologies Ltd), using pre-trained deep neural network model (DiscovAIR Segmentation, v.1.4), which provided precise delineation of individual cells in the imaging data. Approximately $10^4$-$10^5$ nuclei were segmented per sample. To ensure comprehensive coverage of each cell area, we expanded the cell masks by two micrometers. The critical task of assigning reads to cells was accomplished using Baysor[21]. Taking the segmentation mask generated with BIAS and the location of every decoded read as an input, Baysor enabled us to efficiently associate the sequencing reads with their respective cells, facilitating downstream analyses. By implementing these steps, we were able to establish a robust and reliable cell-to-read assignment pipeline for our research, providing a solid foundation for further investigations and insights into the biological processes under study.

**Clustering and subclustering method.** With the aim of identifying cell populations present in the tissue, cell-by-gene matrices from different regions and donors were pooled together. Cells with less than 3 genes and less than 5 transcripts per gene were excluded from analysis. The counts of the remaining cells were then normalized and log-transformed prior to clustering. Main clusters identified were further subclustered, occasionally with excluding 1-2 non-marker genes which interfered with the clustering in order to guarantee the division of cells based on their cell identity. All details can be found in Supplementary Data 3.

**Cell type annotation.** Manual doublet exclusion was performed for each cluster based on scRNA-seq data and previously published annotations[1]. Gene detection levels ranged from 1 to 20 dots per cell (each dot representing mRNA molecule), consistent with the differences in the expected gene expression levels[1]. Approximately 1-2% of misannotation was observed, allowing few AT1, AT2 and aerocyte-annotated cells to be mapped to tracheal regions, some SMG cells to the parenchymal regions, and stromal cells to epithelium. For overall analysis in the current study this misannotation was insignificant, but it should be considered for refined cell type mapping using our HybISS-based atlas. Additional validation was used to define prediction scores using integration mapping method[22] with level 2 annotations from

scRNA-seq[1] as a reference dataset (https://satijalab.org/seurat/articles/integration_mapping). The dictionary of cell types was created to achieve matching cell type groups (Supplementary Data 6) and the analysis files are available at https://www.ebi.ac.uk/biostudies/studies/S-BSST2189. Annotated cell types can be found in the web data viewer (https://adult-lung-iss.serve.scilifelab.se/), which was created using TissuUmaps[44].

**Measurement of baso-luminal cell type distribution.** With the aim of assessing the baso-luminal gradients present in the human lung airways, we first manually segmented individual airways based on DAPI and complementary H&E stainings, keeping only cells detected within the epithelium for further analysis. Next, we manually defined the basal layer using TissUUmaps and we computed the minimum distance from each transcript and cell detected with ISS to the basal layer. Finally, since airways present a different width along the proximodistal axis of the lung, we normalized the distance to the basal layer of each cell/transcript per airway, which resulted in each transcript/cell presenting a relative proximity to the basal layer (0-1). In this context, the closest transcripts/cells to the basal layer will have a distance of 0, whereas the most distant ones will present a relative distance of 1. In order to statistically define basal, luminal and intermediate-enriched genes, we have binned the distances into 10 groups and calculated the mean expression per distance. We used trachea as region with thickest epithelium. The top expression level (>1500 dots by sample) genes that appeared in all four donors were combined, and analysis of variance was performed to define mean distance differences. Genes with lowest variance (<0.15) between donors were selected and tested for highest variability between layers (Std>0.05). The genes returned as a result were combined into the expression distribution plot. Significant differences between genes were calculated using nested one-way ANOVA followed by Tukey's multiple comparisons test.

## Scrinshot

**Probe design.** All padlock probes for SCRINSHOT were designed with a unique barcode, as described previously https://github.com/AlexSount/SCRINSHOT_scripts_for_EMBO_course/blob/main/padlock_probe_design_v6_empty_stable_backbone[32] as well as with new approaches described here: https://github.com/alexandra-firsova/Barcode-design-of-padlock-probes/ and here: https://github.com/HelmholtzAI-Consultants-Munich/oligo-designer-toolsuite. All approaches demonstrated similar results and were not distinguished in results. In order to detect five genes per hybridization cycle using DAPI for nuclei, detection probe oligos were conjugated to one of the five different fluorophores: FITC, Cy3, TexasRed, Cy5 and Cy7. List of SCRINSHOT probes is available in Supplementary Data 7. Probes (padlock and detection) were produced by Integrated DNA Technologies (IDT) and are available to be shared by contacting the corresponding author.

**Probe application and detection.** Experimental procedure in SCRINSHOT was performed as previously described[16,20,32]. Briefly, tissue sections were fixed for 10 minutes in 4% paraformaldehyde, treated with 1 M HCl, blocked and incubated with padlock probe mix. Probes of highly expressed genes were reduced to 1-2 padlocks per gene, the rest was applied at 3-4 padlocks per gene. Ligation with SplintR, and probe amplification steps were followed by fixation of cDNA product and detection cycles. Detection probes were applied at 30 °C in 30% formamide solution, followed by washes at 30 °C in 20% formamide solution. All samples were processed with cell type panel (Supplementary Data 8) in three experiments with variation in probe list for *ZG16B, GRP, ZFP36L2, NKX2.1, CD69* and *BPIFB1*, and slight changes in concentration of padlocks for highly expressed genes. Rare cell type panel was composed of the epithelial cell type markers and rare cell type markers (Supplementary Data 8). It was applied to samples from

four donors, but only two samples per location were selected for further analysis (Supplementary Table 1). Additionally, *SERPINB3 and KRT13* were detected in samples from donors 1 and 4 together with other epithelial markers. Images were taken at 20x magnification using as a Z-stack with 10-11 steps of 0.8 µm (to cover the whole 10 µm thickness) at a widefield microscope (Zeiss Axio Observer Z.2, Carl Zeiss Microscopy GmbH, with a Colibri led light source, equipped with a Zeiss AxioCam 506 Mono digital camera and an automated stage).

**Data analysis.** Analysis of SCRISNHOT data was performed as described previously[16,20,32]. Projection and stitching, followed by image export was performed in Zen (2.3 lite). Tiling of SCRINSHOT images was performed in Fiji (ImageJ1.53c), SCRINSHOT signal (dot) detection was done in CellProfiler (3.1.9). Automated nuclei segmentation was performed in BIAS on tiled images (via Image Filters function) using deep neural network model (DiscovAIR Segmentation, v.1.0) in Segmentation function with scaling 1.50-2.00, detection confidence 1%, contour confidence 50%. Manual correction of segmented nuclei shapes in areas with compact tissue, such as bronchial epithelium, was used via Manual Segmentation function. Approximately $10^4$-$10^5$ nuclei were segmented per sample. Nuclei shapes were then expanded by 0.5 µm using Mask Operators function, overlapping shapes were removed at tile borders using Remove Duplicate function at 10% overlap threshold, then nuclei shapes were eroded back by 0.5 µm in order to create a gap between regions of interest using Mask Operators function. Images of nuclei masks were exported using Scan function, and used to define regions of interest (ROIs) in Fiji. Nuclei regions were expanded by 2 µm (without overlaps) to recapitulate cell ROIs using CellProfiler. Assignment of dots to cells was performed in Fiji, as described previously[16,32]. Cell type panel data was used for clustering for cell type confirmation, epithelial cell quantification, epithelial cell type mapping, SMG mapping, muscle and fibroblast gene expression level comparison, and immune cell mapping. Rare cell panel was used for manual annotation and quantification of rare cells. COPD panel was used for selected cell type annotation, as in Supplementary Data 5, quantification of cell types and neighborhood enrichment analysis.

**Clustering data and mapping cell types.** Gene detection levels ranged from 1 to above 50 dots per cell (each dot representing mRNA molecule), consistent with the differences in the expected gene expression levels[1,20]. SCRINSHOT analysis code can be obtained here: https://github.com/alexandra-firsova/Cell-analysis-in-SCRINSHOT, https://doi.org/10.5281/zenodo.17167875. For cell type clustering with 64 genes, negative and low positive cells were excluded by filtering out cells with less than 25 total dot count, leaving around 40% of cells for further analysis. For a shortened cell type panel of 41 genes used for COPD analysis the filtering of low positive cells was performed by excluding cells with less than eight counts per cell. Normalization by total count was the only data conversion applied. In order to avoid sample-specific cluster separation in epithelial and immune cells (Supplementary Fig. 4) and to achieve finer cell state separation, data was clustered separately for each sample, and uniform annotation guidelines were applied (Supplementary Data 5). For comparison of healthy and COPD samples, datasets were combined. Different resolutions and principle components (pc) were tested on a representative dataset, so that most expected cell types appeared as clusters. Leiden clustering with 20 nearest neighbors, resolution 1.5 and pc number 7 was used for all remaining datasets. Subclustering was performed for selected clusters to retrieve smaller groups of cells with 10 nearest neighbors, resolution 0.4 − 1.5 and pc number 0. Clusters were manually annotated according to expected marker genes. Manual cell type assignment for rare cell types was performed by positivity (>20% of maximum counts per cell for each gene) for any of the rare cell type

marker genes. The rare cells were then clustered as described above. Annotated cell types were mapped by ROI coordinates using TissuUmaps[44], and can be found in the web data viewers.

**Airway length measurement.** Airway length or circumference was measured by drawing a line along the basal membrane (based on the border of KRT5 and KRT15 signal) in Fiji and measuring its length in micrometers.

**Analysis of cell states in histological tissue compartments.** Histological tissue compartments were assigned manually on the tissue sections stained with hematoxylin and eosin after SCRINSHOT. The assigned regions were chosen to include the large airways, terminal and respiratory bronchioles, large and medium blood vessels, cartilage, submucosal gland and alveoli. The compartments included their surrounding connective tissue. Guided by the presence of characteristic epithelial cells, we defined the SMG and peri-SMG mesenchyme by selecting the tubular structures located between the airway epithelium and the cartilage, and their surrounding connective tissue (usually 50-100 µm from the basal membrane of the tubular structures). The airways and peri-bronchial compartment, which was thick in the trachea (up to 400 µm) and thinner in distal airways (100-200 µm), was defined by subepithelial mesenchymal cells, smooth muscle fibers and connective tissue. The alveolar compartments were defined by alveolar structures, which were not in direct contact with large vessels or airways. The remaining histologic regions lacked epithelial structures and were distinguished either by the presence of large vessels or cartilage structures. Vessel compartments were divided into peri-venous and peri-arterial, according to the histology of the surrounding mesenchyme (including smooth muscle layer or tunica adventitia), which is usually thicker (up to 300 µm) in the arteries than in veins (up to 100 µm). The peri-chondrial compartment included cartilage and its surrounding peri-chondrial connective tissue (extending up to 100 µm). The proportions of non-epithelial cells within these compartments were quantified according to their clusters and statistically compared between the anatomic location, donors and compartments using PERMANOVA statistical test (R). For the statistical analysis of peri-epithelial compartments, the proportions of the non-epithelial cell types/states per compartment per region per donor were used. In case if multiple structures were present in the sample, the data was combined per donor. Therefore each cell type/state proportion in each of the peri-epithelial compartments gave one value per donor. Initially, the comparison of all compartments simultaneously was performed for each cell type using Friedman test followed by Dunn's multiple comparisons test. To complement that, the pairwise comparisons of these proportions were calculated for all cell types/states for each anatomical region separately, using the paired Wilcoxon ranked test. Kruskal-Wallis test was used for multiple compartment comparison for each cell type in non-epithelial compartments due to lack of donor matching per compartment.

**Visium and RNA-Rescue Spatial Transcriptomics (RRST)**
Visium and RRST were performed as described previously[17]. Proximal and distal lung samples from donor 4 was used for standard Visium in 8 sections 100 µm apart from each other, and in a single section of trachea from the same donor. RRST was used in proximal and distal lung sections from donor 4 and distal lung section from donor 1. Cell type annotation was performed using stereoscope[23]. For Visium sequenced libraries were processed using Space Ranger software (version 1.2.1 for standard Visium data and version 1.3.1 for RRST data, 10× Genomics). Analysis was carried out using R. Custom code and data is available at https://github.com/ludvigla/DiscovAir_data_explorer/ (v1.0) and https://github.com/ludvigla/RRST (v1.0.0). Graph panels were created using viewer app and the annotated dataset available online: https://github.com/ludvigla/DiscovAir_data_explorer/.

## Correlation analysis between three methods

Serial sections from a distal lung sample (donor 1) were used for manual annotation of histological compartment regions. Three characteristic compartments (a distal airway, an artery and alveolar tissue) were selected and cell type proportions within these compartments were analysed. For SCRINSHOT and HybISS percentage of detected cell types was defined against all segmented nuclei. For Visium the Stereoscope results (0-1 values per spot) from RRST analysis were used, top 10 detected cell type values per spot (diameter: 55 μm) were included in the analysis, and multiplied by 10 to get the estimated cell numbers. Total cell number per area was considered the number of spots multiplied by 10 (each spot containing approximately 10 cells). In order to match cell type annotations between methods, all cell types and subtypes were grouped to the lower level of annotation to create 17 cell types (for example, all vascular and capillary cells were grouped into a category 'Endothelial', ' Immune nan' contained B plasma, B lymphocytes and DCs, which could not be confidently annotated in SCRINSHOT): endothelial, fibroblasts, smooth muscle, T and NK (summed), immune other, AT1, AT2, basal, ciliated, macrophages, mast, monocytes, mucous, SMG intermediate (or duct), suprabasal, secretory, serous. Cell type proportions were assigned as X and Y values on the plots, and three methods were compared pairwise, using Pearson and Spearman correlation analysis. Epithelial nan, deuterosomal and myoepithelial cells (not detected in Visium) were also included in SCRINSHOT to HybISS correlation analysis in other anatomic locations.

## Neighborhood enrichment analysis

To explore the cellular environment of each cell and define cellular niches, each cell was re-defined based on the local neighborhood of each profiled cell. For each cell, its 20 closest cells were considered its microenvironment and used to create a cell-by-neighboring cell types matrix. For every cell, we quantified the amount of cells of each cell type present in its defined neighborhood, as done previously[37]. Cell-by-neighborhood matrices were then preprocessed following standard single-cell preprocessing steps including library size-based normalization. To define cellular neighborhoods in COPD vs healthy dataset, graph-based clustering was performed using Leiden clustering. Neighborhoods were further represented via UMAP low dimensional representation. Clusters resulting from this process represent tissue neighborhoods, defined as groups of cells that present the same local microenvironment. Since some cellular neighborhoods were sample, or even cell specific, local clusters with less than 50 cells were excluded from the analysis, as they did not represent general neighborhoods, but rather unique rare microenvironments.

## Immunofluorescence

Immunofluorescent staining was performed on serial fresh-frozen lung sections from healthy atlas cohort. After sample drying, fixation was performed with 4% paraformaldehyde for 10 minutes, followed by blocking in 5% donkey serum and 0.1% triton diluted in phosphate buffer (PBS) for 30 minutes at room temperature. The following antibodies were applied overnight at 4 °C: anti-GHRL rat monoclonal antibody (R&D Systems, MAB8200-SP, clone: 883622, wd:1.25 ug/ml, Lot: CILU0220021), anti-GRP rabbit polyclonal antibody (Bioss, bs-0011R, wd:1:200, Lot: AI08112480), anti-E-Cadherin mouse Alexa Fluor 555 (BD Biosciences, 560064, clone: 36/E-Cadherin, wd: 1:100, Lot: 4337645), anti-alpha Smooth Muscle Actin (ACTA2) eFluor660 mouse monoclonal antibody (Thermo, 50-9760-82, clone 1A4, wd:1:100, Lot: 4347892), anti-SPARCL1 goat polyclonal antibody, (BioTechne, R&D, A2728, wd: 2 ug/ml, Lot: VHT0323061), anti-CLDN5 Alexa Fluor 488 mouse monoclonal antibody, (Thermo, 352588, clone 4C3C2, wd: 1:100, Lot: UF285712). After three washes secondary antibodies from Jackson ImmunoResearch were applied for one hour at room temperature, followed by DAPI counterstain. Imaging was performed as for SCRINSHOT. Images are available as supplementary data.

## Reproducibility and statistical analysis

For all experiments biological replicates were performed as follows: four for HybISS, two for Visium and RRST (distal lung only), three for SCRINSHOT trachea and four for SCRINSHOT proximal and distal lung (for each marker panel), three for healthy and COPD, four for endothelial marker immunofluorescence and three for rare cell immunofluorescence. Data normality for percentage values was reached using logit-transformation. Variance was tested in using F test. Statistical tests for each corresponding dataset are indicated in figure legends or method description. When normality could not be reached (for example, 0-inflated data), non-parametric tests were used for data comparison. For too low/undetectable values the 0 were replaced half of the minimum detection values. Statistic comparisons were performed and graphs were created in GraphPad Prism 8.3.0 (538), or in Microsoft Excel (14.0.7268.5000).

## Reporting summary

Further information on research design is available in the Nature Portfolio Reporting Summary linked to this article.

## Data availability

The HybISS data and analysis generated in this study have been deposited in the github database (https://github.com/Moldia/adult_lung_DiscovAIR_spatial) and under https://doi.org/10.5281/zenodo.17175597 [https://zenodo.org/records/17175597]. The raw SCRINSHOT data images were deposited in BioImage Archive under accession codes S-BIAD2307 for healthy lung atlas, S-BIAD2308 for rare cells, and S-BIAD2310 for COPD cohort. The Visium and RNA-Rescue (RRST) data are available at https://github.com/ludvigla/DiscovAir_data_explorer/ (commit db99c5d, Release v1.0). All processed data (SCRINSHOT and HybISS) and cell type map files, gene expression matrices and coordinates, immunofluorescent and histological images are available with accession number S-BSST2188. The statistical data generated in this study are provided in the Source Data files and at https://www.ebi.ac.uk/biostudies/studies/S-BSST2189. The interactive viewers of cell types and links to all data are available under the following link: https://github.com/alexandra-firsova/Human-lung-cell-atlas (and https://zenodo.org/records/17179727). Source data are provided with this paper.

## Code availability

SCRINSHOT analysis example is available under this link: https://github.com/alexandra-firsova/Cell-analysis-in-SCRINSHOT, https://doi.org/10.5281/zenodo.17167875 [https://doi.org/10.5281/zenodo.17167875]. SCRINSHOT probe design: https://github.com/alexandra-firsova/Barcode-design-of-padlock-probes/, https://doi.org/10.5281/zenodo.17171960 [https://doi.org/10.5281/zenodo.17171960]. Visium-based analysis: https://github.com/ludvigla/DiscovAir_data_explorer/ (commit db99c5d, Release v1.0). HybISS analysis: https://github.com/Moldia/adult_lung_DiscovAIR_spatial, https://doi.org/10.5281/zenodo.17175597 [https://doi.org/10.5281/zenodo.17175597].

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

## Acknowledgements

We acknowledge discovAIR grant agreement 874656 funding all authors of this manuscript, and all members of discovAIR consortium, especially Karen van Eunen, Pascal Barbry, Laure-Emmanuelle Zaragosi, Amanda Oliver and Elo Madissoon. We gratefully acknowledge the provision of human biomaterial and clinical data from the CPC-M bioArchive and its partners at the Asklepios Biobank Gauting, the Klini-kum der Universität München and the Ludwig-Maximilians-Universität München, especially Anja Disovic and Marion Frankenberger. We are grateful to the lung tissue and organ donors and their families for the gift of human tissue, and we thank the patients and their families for their support. We also kindly acknowledge the ISS facility in Science for Life Laboratories, Stockholm, Sweden, including Chika Yokota and Amitha Raman, for performing the HybISS experiments. Finally, we acknowl-edge Jan-Olov Persson at The Statistical research group (SFG), Depart-ment of Mathematics, Stockholm University, for consulting in statistical data analysis. FK and PH acknowledge support from the TKP2021-EGA09, Horizon-BIALYMPH, Horizon-SYMMETRY, Horizon-SWEEPICS, Horizon-Fair-CHARM, HAS-NAP3, OTKA-SNN no. 139455/ARRS, and Finnish Cancer Society. This work was supported by the EU Horizon

Program (DiscovAIR). MN acknowledges Cancerfonden grant CAN 2021/1726. MCN was supported by the Chan Zuckerberg Initiative, LLC Seed Network grant CZF2019-002438 "Lung Cell Atlas 1.0". CS laboratory is supported by VR 2019-04893 from Cancerfonden 21 1794 P = 1H and Erling-Persson Foundation 2023-0035.

## Author contributions
The DiscovAir Consortium (work package 2), led by C.S. and M.C.N., conceived the project. A.B.F. managed and coordinated the project. L.B.K. and M.D.L. designed cell type marker panels using Spapros and analyzed scRNA-seq datasets. L.B.K. assisted with additional marker gene selection and data interpretation. K.T.M. collected and froze the donor tissue samples for the Atlas dataset. M.G.S. provided access of tissues from CPC-M bioArchive. M.G.S., Y.C., J.G.S. collected the tissues and prepared the frozen samples. A.B.F., A.S., A.L., L.B.K. and J.T. designed SCRINSHOT probes, code for probe design provided by A.S. and L.B.K. A.B.F., X.M.A., Z.A. and L.A.G. performed tissue cryosectioning and histological staining. A.B.F. and W.T. performed histological and pathological assessment of sections. X.M.A., Z.A. and L.A.G. performed RIN analysis, Visium and RRST. L.L. analyzed Visium and RRST datasets and performed Stereoscope-based annotations. A.B.F. performed SCRINSHOT experiments and analyzed SCRINSHOT dataset. A.S. assisted with SCRINSHOT image alignment and data visualization preparation. S.M.S. analyzed HybISS dataset provided by ISS facility. T.B. and F.K. optimized the automated nuclei segmentation design, algorithm training dataset of manually-drawn nuclei was provided by A.B.F. and A.S. A.B.F. and S.M.S. performed data clustering and cell type annotations. A.S. performed integration mapping. S.M.S. performed measurements of proximo-distal and baso-luminal cell type and gene expression distribution and neighborhood enrichment analysis in healthy (HybISS) and COPD (SCRINSHOT) datasets. A.B.F. performed method correlation analysis, measurements of rare cell types, annotation and analysis of tissue compartments, and COPD-related cell type proportion changes. A.B.F. and J.T. performed immunofluorescent staining. C.S., M.C.N., K.B.M., M.N., J.L., F.J.T., M.D.L., H.B.S., W.T., J.B. and P.H. supervised the project. A.B.F., S.M.S. and C.S. wrote the manuscript, and all authors reviewed it.

## Funding

## Competing interests
M.D.L. contracted for the Chan Zuckerberg Initiative and received speaker fees from Pfizer and Janssen Pharmaceuticals. J.L. and Z.A. are the scientific consultants for 10x Genomics Inc which holds intellectual property rights to the spatial transcriptomics technology. F.J.T. is a scientific consultant for Immunai Inc., CytoReason Ltd, Cellarity, BioTuring Inc., Genbio.AI Inc., and has an ownership interest in Dermagnostix GmbH and Cellarity. M.N. is co-founder of CARTANA. The remaining authors declare no competing interests.

## Additional information

[1]Science for Life Laboratory, Stockholm, Sweden. [2]Department of Molecular Biosciences, The Wenner-Gren Institute, Stockholm University, Stockholm, Sweden. [3]Department of Biochemistry and Biophysics, Stockholm University, Stockholm, Sweden. [4]Institute of Computational Biology, Helmholtz Munich, Neuherberg, Germany. [5]School of Life Sciences Weihenstephan, Technical University of Munich, Munich, Germany. [6]Department of Gene Technology, Kungliga Tekniska Högskolan (KTH), Stockholm, Sweden. [7]Department of Surgery, University of Cambridge, and Cambridge NIHR Biomedical Research Centre, Cambridge, UK. [8]Synthetic and Systems Biology Unit, HUN-REN Biological Research Centre, Szeged, Hungary. [9]Single-Cell Technologies Ltd, Szeged, Hungary. [10]Institute of AI for Health, Helmholtz Munich, Neuherberg, Germany. [11]Institute for Molecular Medicine Finland, University of Helsinki, Helsinki, Finland. [12]Research Unit for Precision Regenerative Medicine (PRM), Helmholtz Munich, Neuherberg, Germany. [13]Comprehensive Pneumology Center (CPC), Member of the German Center for Lung Research (DZL), Munich, Germany. [14]Department of Rheumatology and Immunology and Department of Pulmonary Medicine, Allergology and Clinical Immunology, Inselspital, Bern University Hospital, University of Bern, Bern, Switzerland. [15]Lung Precision Medicine (LPM), Department for BioMedical Research (DBMR), University of Bern, Bern, Switzerland. [16]Center for Thoracic Surgery Munich, University Hospital of the Ludwig-Maximilians University (LMU), Munich, Germany. [17]Asklepios Medical Center, Gauting, Germany. [18]Institute of Lung Health and Immunity, Helmholtz Munich, Neuherberg, Germany. [19]Department of Medicine V, University Hospital of the Ludwig-Maximilians University (LMU), Member of the German Center for Lung Research (DZL), Munich, Germany. [20]Wellcome Sanger Institute, Wellcome Genome Campus, Cambridge, UK. [21]Department of Pathology and Medical Biology, University of Groningen, University Medical Center Groningen, Groningen, The Netherlands. [22]Groningen Research Institute for Asthma and COPD, University Medical Center Groningen, Groningen, The Netherlands. [23]Institute of Experimental Pneumology, University Hospital of the Ludwig-Maximilians University (LMU), Munich, Germany. [24]School of Computation, Information and Technology, Technical University of Munich, Munich, Germany. [25]Department of Internal Medicine, Molecular Pneumology, ECCPS, Justus Liebig University, Member of the German Center for Lung Research (DZL), Giessen, Germany. ✉e-mail: christos.samakovlis@su.se

