## [Transparent Peer Review file · Nature Communications]

Spatial single cell atlas reveals regional variations in healthy and diseased human lung

Corresponding Author: Professor Christos Samakovlis

Version 0:

Reviewer comments:

Reviewer #1

(Remarks to the Author)

In this study, the authors analysed lung biopsies using various spatial transcriptomics tools and described the cells present in these tissue sections. They also showed different ways to represent the data, characterised rare epithelial and endothelial cells, and finally utilised COPD samples to compare healthy versus diseased tissue. The study is interesting although quite descriptive, presenting some very intriguing data, though the authors could explore certain findings more deeply, given their expertise in the field.

Specific comments

1. The introduction is well-written but concise. Given that Nature Communications is a general journal, it would be helpful for non-specialists in lung research if more details were provided on why COPD was chosen as the disease to analyse.
2. It is essential to provide a better explanation of why the methods (HybISS, SCRINSHOT, RRST) were chosen. Is HybISS published? Why were these particular technologies chosen - RRST - SCRINSHOT - HybISS? In what ways are they complimentary? The exact analysis pipeline is not clear.
3. One major limitation of the study that should be discussed is the selection of "healthy" donors. Three out of four samples were obtained from smokers or ex-smokers, with a vast age range (28 to 74 years). Additional information on the donors, such as the duration and quantity of smoking, should be provided. Do the data from the non-smoker differ from those of the other three donors? How do cells from each donor contribute to the cell types identified?
4. Figure 1: The authors first used HybISS to analyse tissue sections and performed single-cell spatial transcriptomics. HybISS relies on a limited number of markers (i.e. 162), and some of the selected markers are questionable. How were they chosen? For instance, CD4 T cells were characterised by BTG1 and CD52 alone, but these genes are not specific to CD4 T cells according to the authors' own data (<https://5locationslung.cellgeni.sanger.ac.uk/all>).
5. In general, it's not clear how quantitation is achieved? Most of the figures have images, which are attractive but it's not clear how they were chosen and how they relate overall to the data? It is difficult to make such definitive statements about such small sample sizes? In Line 93 - data from only 4 donors included? The authors then state that 14 sections from four donors were chosen – but why 14, how were they selected and where were these sections taken from which area/anatomical region? This information is critical for the interpretation of the data, but also for the statistical analysis – it is not clear in every figure how this analysis is conducted.
6. In Sup fig 2 - no neutrophils were identified? line 116, states that “additional cell type should not be assigned because of low abundance or sparse gene expression” - but is that a facet of the technique (ie the sensitivity) or really low levels of gene expression? This would be hard to believe for T cells? And others have identified even rare cells such as tuft or ionocytes in lung tissue sections. How did this differ in each of the different locations - i.e. could you pick up tuft cells in the tracheal areas but not the proximal/distal lung regions?
7. Much of the data for HybISS and SCRINSHOT are presented together - but why - what does each signify? How are they comparable?

Line 122 - says that RRST was used to "complement the HyBISS data - but why just HyBISS? Not SCRINSHOT? - What does this technique give that the others don't. Supp fig 3 only shows proximal and distal lung - what about the trachea?

8. Figure 2: The description of epithelial cell diversity and the methods for visualising cellular neighbourhoods and gene expression spatially are commendable. However, most of these representations show data from only one donor. The authors need to incorporate data from all donors, or a collective analysis of all donors, into the spatial distribution of genes and cell types? This may provide more informative insights.

5. Line 155 - the authors have labelled cells as "nan -not annotated" – but its not clear why they are not annotated - they represent a relatively large proportion of cells in epithelium - was this true of all techniques? Still seem to have "nan" cells using several techniques as in supp fig 4?

6. Line 207 - conclusion for section - not clear what novelty is here?

7. Fig 3 - section on rare cell types - I think this is potentially very interesting - but how rare is rare? The analysis of rare cell types is a true challenge with these tools. The authors define four groups of neuroendocrine (NE) cells, though the expression of some markers, notably CHGB and PCSK, is very low (i.e. as low as two spots). Additional validation should be provided to confirm the existence of these cells. Ideally, this could be done through protein staining in tissue sections or using other datasets. It would be interesting to make a comparison to other more "common" cell types? For example in relation to basal cells, or goblet or ciliated cells. Overall, the data in fig 3 is difficult to follow. - particularly 3C where the cells are very blurry – the reader needs a better description of what they are trying to depict here... Why does the heatmap only represent two donors rather than all four?

8. Fig 4 relates to the idea of different cell states existing in different tissue compartments, and talks about neighbourhood analysis (line 243) but I'm not clear where this analysis is presented or how it was done?

Figure 4: The regions defined here are well described, and the spatial transcriptomics approach is used effectively to extract meaningful information. However, some aspects are confusing. Where is the data showing the different populations of endothelial cells expressing either SPARCL1 or CLDN5? These populations appear in the heatmap but were not shown earlier. Do they represent distinct clusters within the endothelial cells? They seem to represent a small proportion of the data, and the grey-scale heatmap makes it difficult to visualise. On what type of blood vessels are these cells located? The lung regions contain a vast network of capillaries, post-capillary venules, and small arterioles. It would be valuable to investigate where SPARCL1+ or CLDN5+ cells are located. The authors mention venous or capillary cells, but the images are too zoomed out to confirm these observations. Excellent antibodies are available for these molecules, and it would be crucial to stain lung sections to verify the presence of these markers on endothelial cells at the protein level.

Most of the observations described in this section rely on heatmap representation, and the authors draw strong conclusions from this. Statistical analyses have been conducted, but the asterisks (*) are not clearly visible on the heatmap. Additionally, it is unclear which groups were compared. Was it individual donors or compartments within each donor? In the figure (fig 4) what are the n? Are they regions or patients or sections? If so how do they relate to the tissue sections? One from each or more? If so how were they selected?

From the images, it appears that SPARCL1+ cells are close to T/NK cells. Are these T/NK cells different from the rest? What is the immune cell neighbourhood of the different endothelial cell populations?

8. Figure 5: The final part of the paper focuses on the analysis of COPD samples. Although the sample size is limited, the data presented are potentially interesting, and the methods and visualisations for analysing cellular neighbourhoods are excellent.

The methods section of the patient sample collection is confusing – for the healthy lung cell atlas it is stated that the tissue was taken from deceased organ donors – so these were all healthy, with no previous history of any respiratory conditions? For the COPD and non-COPD samples – line 445, it doesn't state how these patients were chosen? Tissue is taken from resections from cancer patients that had COPD - is this a good choice of tissue? Surely, they are unlikely to be normal at transcript level if from cancer patients, even if the histological assessment is "normal/cancerous"? In the table the donors are n=1, non-COPD and n=3, COPD - these are extremely small numbers to make definitive conclusions about anything. The AT0 cell population is characterised by the co-expression of NAPSA, SFTPC, SCGB3A1, SCGB3A2, and LTF. However, according to Supplementary Fig. 7C, some of these genes are not expressed by 100% of AT0 cells (e.g. LTF and SCGB3A1). How did the authors decide on these markers? Is the AT0 cell population heterogeneous? A visual representation of all AT cells in both healthy and COPD conditions would be helpful for readers.

9. The discussion is succinct and primarily summarises the study's limitations. It would benefit from additional discussion on the roles of rare neuroendocrine cells or specific endothelial cell populations, and their potential roles in lung function and COPD. Spatial transcriptomics has been successfully employed in other organs, and this could be discussed too. I'm not clear what final conclusions were made from the data regarding our understanding of COPD?

Minor

- The images in Figure 1D are too zoomed out, and higher magnification is needed to visualise the cellular level.
- Heatmaps are difficult to interpret in black and white; could they be changed to colour?
- Very difficult to see the yellow staining in Fig 2F - serpinb3 and SCGB1A1

(Remarks to the Author)

In the manuscript, Firsova et al. performed spatial transcriptomics (ST) profiling and analysis and presented two ST atlases of human lung: (1) a healthy lung atlas based on the ST profiling of samples from three representative regions of human lungs (n=4) using three technologies (HybISS, SCRINSHOT, and Visium/RRST) and (2) a COPD lung atlas based on SCRINSHOT profiling of distal lung samples from 3 patients with COPD GOLD stage II and 3 control lung samples. The authors first identified cell type marker gene panels from a previously published scRNA-seq of human lung and then used these panels for HybISS or SCRINSHOT-based ST profiling of healthy and COPD lung samples. Based on the HybISS data, 35 major cell types were identified in the healthy lung atlas. Based on the SCRINSHOT datasets, 23 major cell types were detected in the healthy lung, and 20 major cell types were defined in COPD. In addition to these major scRNA-seq-based cell types, the authors report the identification of intermediate cell states of suprabasal cells and SMG cells, as well as subtypes of neuroendocrine (NE) cells. In particular, they report the detection of GHRL-positive NE cells, a previously reported embryonic/pediatric NE subtype, in adult lungs and found that these cells were present in bronchioles but not in trachea or proximal lung. Using the healthy lung atlas, consistent regional variations in cell types and marker gene expression were identified. The patterns are consistent with existing knowledge and/or recent single-cell findings. The authors also integrated the ST data with cellular morphologies in H&E image and demonstrated the spatial patterns of specific cell types and marker gene expression in distinct tissue compartments.

Using the COPD atlas, changes in cell proportions are reported, including the decrease of AT1 cell proportions and the increase of T lymphocytes and AT0 cell proportions. The authors performed neighborhood analysis and identified three cellular neighborhoods enriched in COPD and two neighborhoods decreased in COPD. They proposed that these neighborhood enrichments and decreases were consistent or related to known COPD phenotypes or recent single cell findings, including the increase of AT0 cells with known inflammatory signatures of COPD.

Overall, the presented ST atlases of healthy and COPD lungs provide an initial resource for the research community for data exploration and understanding of the spatial patterns of marker genes and cellular neighborhoods in distinct tissue regions/compartments. The pipeline for tissue and bioinformatic analyses will be useful with the increasing interest in spatial transcriptomics for the study of the lung.

Major Concerns:

(1) Sample size and individual variations:

Only three COPD and three control lungs were included in the SCRINSHOT profiling of 41 genes for atlas construction and comparative analyses. Differences in age, sex, smoking, treatment regimen, and other patient characteristics limit conclusions for this small number of samples.

Within this small number of samples, COPD-related metadata was highly variant (Supp Table 1). Regarding smoking status: Smoking status varied among COPD patients. Two of the three controls were non-smokers. Regarding sex: all three control patients were male, while two of the three COPD patients were female; one of the three controls was much younger than the COPD patients. Therefore, the small sample size, together with high individual variations in disease-related metadata, raises a major concern regarding the validity and utility of the comparative analysis.

(2) New insights into the disease:

Using the COPD atlas, the authors demonstrated changes in cell proportions and cellular neighborhoods in COPD, which were mostly consistent with or related to known COPD phenotypes in recent single cell RNA findings. It is not clear that new insights into disease pathogenesis have been generated from the present analyses.

(3) AT0 cells in COPD lung:

Based on the analysis of the SCRINSHOT-based topographic lung atlas of COPD and healthy lungs, the present work reported a significant increase in the proportion of AT0 cells in the COPD samples.

According to the spatial visualization plot (Fig. 5B), the spatial locations of AT0 and AT2 cells look similar. Do AT0 and AT2 have similar or different spatial localization patterns in the COPD lung?

AT0 cells were defined as a population of cells co-expressing AT2 (NAPSA, SFTPC) and airway (SCGB3A1, SCGB3A2, LTF) markers. A recent scRNA-seq study (PMID: 35355018) identified an AT0 cell population co-expressing AT2 and airway markers, which were increased in COPD lungs. The study (PMID: 35355018) was not discussed in the "AT0 cell state alteration" section in the present work. What are the differences between the "AT0" in the present work and the one in PMID: 35355018?

(4) GHRL-positive neuroendocrine (GHRL-NE) cells:

GHRL-NE cells were "detected in embryonic and pediatric datasets and were hypothesized to gradually disappear in adulthood" in a previous study. In the present work, GHRL-NE cells were identified in adult lungs and were found to be specifically located in distal bronchioles but not in trachea or proximal lung. Orthogonal experiments that support this conclusion would strengthen the work.

(5) Accuracy of cell type identification:

Cell types in both healthy and COPD lung atlases were identified mainly based on HybISS or SCRINSHOT profiling of pre-defined cell type marker gene panels, which were selected on the basis of published scRNA-seq datasets. These cell types

formed the basis for major analyses in the present work, including cell type proportion analysis, identification of regional differences and variations, neighborhood analysis, and disease alteration analysis. Therefore, it is important to provide assessments of the accuracies of the identified cell types,

In Supp Fig 2, dotplots were used to demonstrate the expression patterns of cell type marker genes in the published scRNA-seq cell types (panel C) and in the present HyBISS cell types (panel D). However, several markers did not exhibit a clear selective pattern in their corresponding HyBISS cell types (Supp Fig. 2D), such as HPGD, secretory cell markers (WFDC2, SCGB1A1, MUC5AC), APOE, DCN, MUC5B, etc.

Were all the annotated HyBISS cell types supported by the selective and/or abundant expression of corresponding marker genes? It would be better if statistical analyses could be performed to quantify to support the accuracies of HyBISS cell-type annotations.

Likewise, visualization and quantification should be performed to support the accuracies of SCRINSHOT-based cell type annotations.

(6) Robustness of cell type annotations and proportions:

In the healthy lung atlas, HyBISS and SCRINSHOT were applied to profile serial sections of the same region, providing an opportunity to assess the robustness of the reported cell type annotations and proportions. In line 128, the authors state that “the location of assigned cell types within the tissue was consistent between all three methods.” However, this conclusion is supported by visualization of subsets of cell type annotations in three regions (Supp Fig. 3). Statistical analyses should be performed to quantify and compare cell type annotations and cell type proportions between HyBISS and SCRINSHOT profiling of the same regions.

(7) Two of the control samples from the healthy lung atlas were also used in the COPD lung atlas and were re-profiled by SCRINSHOT using a gene panel (41 genes) different from the panel (64 genes) used for the healthy lung atlas providing an opportunity to cross-validate the cell type annotations and proportions in these two control samples, which could be important for identifying disease-related alterations.

(8) Suprabasal vs. basal:

A detailed analysis of the spatial patterns of suprabasal cells was presented. S100A2 was used as the primary marker to distinguish suprabasal from basal cells (lines 162 and 192). S100A2 was selected based on the published scRNA-seq dataset. However, in the dotplot of Supp Fig. 2C, expression frequency and abundance in both basal and suprabasal cells in the scRNA-seq data was detected in S100A2 shown in other bronchial epithelial cells.

How reliable and accurate is the separation of suprabasal and basal cells in the HyBISS data based on S100A2 expression?

(9) Data availability:

The ST data are shared through TissUUmaps-based web interfaces. We tried the web interfaces but found they were slow in loading data, for example, gene expression and metadata visualization.

It would be helpful if the data (such as the processed images and expression objects) could be shared for downloading for exploration and secondary data analysis.

Reviewer #3

(Remarks to the Author)

The manuscript describes a lung cell atlas that is spatially resolved using multiple SRT technologies including data from targeted and untargeted assay. The study revealed the locations of the majority of lung cell types including rare cell types, their variability across locations and distinct cellular neighborhoods, and as a demonstration, the usage of the atlas as a reference in the analysis of a COPD dataset. In general, the manuscript appears well written with substantial details and some novel observations.

I do have several friendly concerns:

1. The atlas is based on very small sample size (4 donors). The complete information about the samples (including age, gender, ethnicity, etc.) need to be disclosed and potential biases need to be discussed?
2. The study used 3 SRT technologies, including 2 targeted: SCRINSHOT and HyBISS as single-cell resolution, and 1 unbiased: Visium at spot resolution. Potential limitation/biases by the technologies need to be discussed and compatibility with other new (commercially available, and/or widely used) technologies such as Xenium, MERFISH, Stereo-seq, etc. needs to be clarified. E.g., how much overlap there is with the Xenium lung panel? How do others use the data from this study as a reference atlas when their data are produced by different technologies?
3. How will the atlas be disseminated? It is great to have a web-based viewer? However, that is clearly insufficient, as it does not allow other users to integrate the entirety of the data as a reference into their studies.
4. The data analysis performed looks largely reasonable (and clear) as described. However, to enable full reproducibility, the code/scripts used in generating the atlas and in performing the COPD analysis (e.g., Neighborhood enrichment analysis) need to be released publicly.

Minor:

- Fig. 5A, can the asterisk for AT0, be moved near the “AT0” label to be consistent with other cell types?

Reviewer #4

(Remarks to the Author)

Reviewer #5

(Remarks to the Author)

Version 1:

Reviewer comments:

Reviewer #1

(Remarks to the Author)

The authors have performed significant experiments to address my queries and have responded thoroughly to my comments.

However, it is important that the authors add a sentence at the beginning of the results section to state that of the four donors used for all of the subsequent analysis 3 were smokers. This really isn't clear from the text, and needs to be.

Reviewer #2

(Remarks to the Author)

The authors have carefully addressed the specific concerns of each of the reviewers. They have clarified the methods, added the requested experimental data, described the limitations, and included appropriate references. All reviewers were concerned about the limited number and heterogeneity of the COPD and control subjects. This issue was not changed in the revision. The comparison, therefore, adds only modestly to current knowledge regarding anatomical or gene expression changes in COPD. Findings are consistent with known aspects of human lung structure and the pathological findings in COPD. The primary value of this work is methodological, providing a comparison of three spatial transcriptomics approaches. The study demonstrates useful cell markers and proximal-peripheral differences in cell types and gene expression patterns. In spite of patient and control heterogeneity, findings are generally consistent across all samples. Statistical approaches, probe sets, cell markers, and availability and accessibility of the dataset will be useful to the field.

In Summary:

Spatial transcriptomics methods and associated analytical frameworks are actively being developed by many laboratories, with the hope that they will yield novel mechanistic insights into the pathogenesis of lung disease.

Reviewer #4

(Remarks to the Author)

RESPONSES TO REVIEWERS' COMMENTS

We thank the reviewers for their support with constructive criticism and concrete suggestions. Below, we address each comment (citation of comments in black font) starting with reviewer 1. We include figures and new analysis in the responses (blue font) and note the corresponding changes in the revised manuscript (blue font, italics). In the revised version, we have added new panels to 13 figures and modified 8 figures to include additional experiments and data analysis. We look forward to the reviewers' responses to our revisions.

Reviewer #1 (Remarks to the Author):

In this study, the authors analysed lung biopsies using various spatial transcriptomics tools and described the cells present in these tissue sections. They also showed different ways to represent the data, characterised rare epithelial and endothelial cells, and finally utilised COPD samples to compare healthy versus diseased tissue. The study is interesting although quite descriptive, presenting some very intriguing data, though the authors could explore certain findings more deeply, given their expertise in the field.

We deeply appreciate the recognition of our findings and the tools utilized for this.

Specific comments

1. The introduction is well-written but concise. Given that Nature Communications is a general journal, it would be helpful for non-specialists in lung research if more details were provided on why COPD was chosen as the disease to analyse.

We acknowledge the comment and have expanded our introduction to clarify our choice of disease as follows:

Finally, we utilized diseased samples available to the consortium from patients diagnosed with chronic obstructive pulmonary disease (COPD), a serious lung disease with globally increasing incidence. This disease is characterized by damage of the distal parts of the respiratory system, such as small airways and alveoli, due to the long-term exposure to toxic particles and gases [18]. COPD samples provide an opportunity to investigate the utility of the healthy atlas in defining aberrant disease-associated cellular neighborhoods, affected by the immune cell composition of the lung, as well as tissue remodeling.

2. It is essential to provide a better explanation of why the methods (HyBISS, SCRINSHOT, RRST) were chosen.

Is HyBISS published? Why were these particular technologies chosen - RRST - SCRINSHOT - HyBISS? In what ways are they complimentary? The exact analysis pipeline is not clear.

We apologize for not conveying the choice of techniques clearly. To address this, we have added a few clarifying phrases to the introduction and restructured the results accordingly in

the first section. The first publication of HybISS as a method is cited in the introduction text, when HybISS is first mentioned (Gyllborg et al). HybISS and SCRINSHOT are both padlock probe-based in situ mRNA detection imaging method, with the differences in the initial steps of probe binding and the last detection steps. HybISS and SCRINSHOT are complementary in their sensitivity (which is higher in SCRINSHOT) and multiplexity (achieved by barcode reading in HybISS). Visium is an untargeted sequencing-based method of mRNA detection, which gives unbiased information of gene expression with lower spatial resolution.

Introduction:

Here, we generated a representative topographic atlas of the healthy adult lung by combining sections from distinct anatomic locations of the respiratory system from four donors of different age and gender. We used three different, multiplexed spatially-resolved transcriptomic (SRT) approaches to obtain complementary results. This included two targeted imaging-based methods with cellular resolution: a highly-multilex HybISS [1] to target the majority of cell types, and a highly-sensitive SCRINSHOT [2] in order to detect a more limited number of cell types and states by variations in gene expression levels. We also employed an unbiased method of mRNA detection with lower spatial resolution (Visium and RRST) [3] to confirm cell types and regional gene expression patterns. This combinatorial approach allowed to deeply characterize consistent, location-related gene expression variability within and across cell types.

We agree that the analysis pipeline was not sufficiently explained in the text for the reader's convenience. The following summary was added to the end of the first paragraph of the results section, after the description of sample selection:

We applied this panel on two selected representative samples using HybISS [1]. The obtained dataset was used to detect most cell types and states in the trachea and lung. In order to detect particular cell states, we used SCRINSHOT, a more sensitive SRT method [2], with additional panels of major cell type markers and genes showing intra-cluster gene expression variation [20]. Finally, a section from each anatomic location was analyzed with an untargeted SRT method Visium in order to validate the probe panels. Visual cross-validation and statistical correlation analysis of the results of the three methods on serial tissue sections, demonstrated consistent marker gene expression patterns, and therefore the performance of targeted methods (Suppl. Fig. 1C).

In addition, after cell type annotation we have included the following paragraph:

Complementing the HybISS datasets, we profiled sequential sections by a SCRINSHOT [16], employing a pre-validated gene panel of 64 marker genes [20]. This panel overlapped with HybISS by 43 genes. The gene counts per cell were obtained as for HybISS, and after clustering major cell types were assigned according to marker gene positivity (Suppl. Table 6). We annotated 24 cell types by clustering, and also defined an additional cell state (AT0) based on previous publications reporting the co-expression of airway and alveolar epithelial markers [9] (Suppl. Fig 4). Two of the cell clusters (immune nan and epithelial nan) could not be further subdivided due to the detection of only general and not discriminative markers. Seven cell types contained multiple clusters or subclusters that were annotated as subtypes or states (Suppl.

Figure 5). We inspected and confirmed the signals and cell types on the tissue to ensure their expected location. We correlated spatial distribution of cell types in HyBISS and SCRINSHOT by plotting them on serial sections and comparing their proportions in the same tissue compartments of serial sections (Suppl. Fig. 6). In addition, we analyzed sequential sections of two tissue blocks using the Visium protocol, followed by the Stereoscope method [23] to deconvolve the cell type composition of each spot using the finest annotation from scRNA-seq dataset from Madisson et al [1] as a reference. Visual assessment confirmed that the location of most assigned cell types within the tissue was consistent between all three methods (Suppl. Fig. 7, A-B). Overall cell type proportion distributions across manually defined tissue compartments on serial sections correlated significantly between the three SRT methods (Suppl. Fig. 7C), confirming the specificity of each of the technologies. The combinatorial approach with three methodologies hand-in-hand overcomes limitations of individual spatial mapping protocols, such as cellular resolution or variable detection sensitivity.

3. One major limitation of the study that should be discussed is the selection of "healthy" donors.

We added a short paragraph in the discussion explaining the limitations of the study: *A limitation of our study is the lack of statistical analysis relating to the variability in age, sex and smoking status of the donors. Despite the limited donor number and including individuals with different physiological status, we observed consistent cell types, states and regional specific cellular neighborhoods in all of them.*

Three out of four samples were obtained from smokers or ex-smokers, with a vast age range (28 to 74 years). Additional information on the donors, such as the duration and quantity of smoking, should be provided.

The requested additional available information about donor health and smoking duration and status is included in the supplementary table 1:

Atlas donor number	Health status	Donor ID	Gender	Age	Smoking status
1	Worsening memory; possible rheumatological pains	640	Female	74	smoked but quit 40 years ago
2	History of kidney and testicular cancer (10 years clear); nothing else of value to report	588	Male	61	non-smoker
3	Thrombocytopenia; liver clinic because of high alcohol intake; has previously had drug overdose. No other known diseases or infections.	689	Male	41	20 cig/day for 21 years
4	Nothing of value to report	583	Male	28	2 cig/day for 10 years

Relevant information from supplementary table 1 including the donor health status and known smoking duration. Cig – cigarettes.

Do the data from the non-smoker differ from those of the other three donors?

This is an important, yet challenging question. According to the integrated healthy lung cell atlas, smoking correlates with little variability in cell type composition, but rather affects gene expression levels within cell types [4]. These smoking-associated genes were not included in the probe panels for spatial analysis. Therefore, any variability in gene expression between the samples cannot be correlated to smoking status or to smoking-associated gene expression signature in the samples. All samples were evaluated by a pathologist using H&E stained sections for the presence of abnormal histologic parameters, and donors with abnormal lung histology were excluded from our healthy cohort. The four donors were selected because their samples had healthy lung histology. According to the cell type presence, no significant differences were observed between donors, and the neighborhood analysis detected cells from all four healthy donors presented in the same regional clusters (Suppl. Fig. 15C). To conclude, the non-smoker (donor 2) had overall similar cell types and their locations, and did not demonstrate any outstanding features.

How do cells from each donor contribute to the cell types identified?

To address this alternative way of looking at cell types, which was not captured in our cell type proportion description, we have reanalyzed the data from all three locations and all samples by donor. Below we show the plots of the proportions of cells from each donor in every cell type identified by HybISS. Of the 35 cell types, 34 were detected in all donors. The rare type of tuft-like cells was detected in 3 out of 4 four donors. We note that tuft cells were detected in all 4 donors with the rare cell type panel and SCRINSHOT. The contribution of cells from each donor to a particular cell type is variable and depends on the size of the captured tissue region in each sample. For example, cells from donor 1 tend to contribute the most to the bronchial epithelial cell types, since the proportion of captured bronchial tissue in donor 1 samples was relatively large. This donor also has the largest proportion of immune cells (24.9% from total annotated cells compared to 8.6-12.9% in other donors), which is indicated by its contribution to the immune cell types.

Parts of whole graphs representing the proportion of cells from each donor per cell type (continued on the next page)

Parts of whole graphs representing the proportion of cells from each donor per cell type (continued from previous page). All 35 cell types are presented, as well as T/NK cell state.

4. Figure 1: The authors first used HybISS to analyse tissue sections and performed single-cell spatial transcriptomics. HybISS relies on a limited number of markers (i.e. 162), and some of the selected markers are questionable. How were they chosen? For instance, CD4 T cells were characterised by *BTG1* and *CD52* alone, but these genes are not specific to CD4 T cells according to the authors' own data (<https://5locationslung.cellgeni.sanger.ac.uk/all>).

The probe selection was performed using Spapros and is based on both positive and negative expression of marker genes [5]. For example, CD4 cells were targeted by a combination of markers (positivity to *CD3* genes, negativity to *CD8A* and higher levels of *BTG1* and *CD52*). This combination was expected to give conclusive results, given that it is difficult to detect *CD4* by mRNA [6]. Since the HybISS-detected expression of these genes on tissue gave sparse results, we did not annotate CD4 cells, but instead kept a general T lymphocyte category, composed of two subgroups: annotated CD8A-positive T cells, and CD8A-negative other/nan T cells. These cell types were annotated manually by positivity to the cell type marker *CD3D* and negativity for NK cell markers, as described in supplementary tables 4 and 5.

To complement the deficits of probe selection and detection by HybISS we used SCRINSHOT with different markers, and detected higher numbers of T lymphocytes that formed a cluster with high *IL7R* expression, and co-expression of *CCL5* (Suppl. Fig 4E).

Balloon plot of SCRINSHOT data demonstrating T lymphocyte cells expressing characteristic markers.

In response to reviewers comment, we additionally performed immunofluorescent staining for CD3 (general T cell marker), CD4, and CD8 in order to confirm their presence in the lung:

Immunofluorescent staining for (left) CD3 (green) on nuclei (DAPI, blue) and (right) CD8 (magenta) and CD4 (yellow) on nuclei (DAPI, white) in a distal lung bronchiole.

These results indicate that *IL7R* probe and immunofluorescent staining are more efficient in detecting T cell subtypes than the HybISS panel of probes used in this study.

5. In general, it's not clear how quantitation is achieved? Most of the figures have images, which are attractive but it's not clear how they were chosen and how they relate overall to the data? It is difficult to make such definitive statements about such small sample sizes?

We apologize that the methods section on data quantification was unclear. In some figures we highlight particular cell types and their niches without showing quantifications. These were detected in all samples and these figures do not make a quantitative argument. They demonstrate cell positions and spatial relationships. The images of cell types are used to represent the statements and to achieve a richer visualization compared to statistical graphs. Below we clarify how the data was processed, quantified and selected for presentation:

1. For HybISS and SCRINSHOT all gene counts per cell were log-normalized and clustered, and clusters were annotated according to the marker genes. Percentage of cell type numbers per region was quantified. Cellular neighborhoods were analysed using neighborhood enrichment analysis [7]. For HybISS clusters, cell type distribution and cell colocalisations are shown in Figure 1. For SCRINSHOT, clusters and subclusters are now added to Suppl. Fig. 4, 5, 12 and 13. Cell type quantifications in histological compartments are shown in Fig. 2A and D and Suppl. Fig. 8E. Epithelial cell quantifications are shown in Suppl. Fig. 11, Suppl. Fig. 12C, and in Fig. 4D. SCRINSHOT neighborhood analysis is shown in Fig. 5 and Suppl. Fig. 13.
2. We present additional quantifications: 1) integration analysis between the HybISS and scRNA-seq data in Suppl. Fig. 3A. 2) correlation analysis of the three methods in Suppl. Fig. 7C, and additional correlations between SCRINSHOT and HybISS in Suppl. Fig. 6B. 3)

To address the ambiguity of quantification description, we also added the reference to quantification to each figure legend.

In Line 93 - data from only 4 donors included? The authors then state that 14 sections from four donors were chosen – but why 14, how were they selected and where were these sections taken from which area/anatomical region? This information is critical for the interpretation of the data, but also for the statistical analysis – it is not clear in every figure how this analysis is conducted.

The aim was to select representative samples from each of the three anatomic locations: trachea, proximal lung and distal lung. Therefore, we created data from at least 12 samples (from three locations of the four donors). For reproducibility we selected 2 additional distal lung samples from a different lobe (for donors 2 and 3) and processed them with HybISS. For SCRINSHOT analyzed 11 samples, because we excluded one tracheal section (donor 3) due to suboptimal RNA quality of the sample. We have clarified this in the text, adding sample number information to Methods 1.2:

In order to create representative data from each of the three anatomic locations, we collected three locations from four donors (a total of 12 regions). For HybISS with additional distal lung samples from a different lobe (for donors 2 and 3). For donor 2 and 3 additional distal lung

samples were used. For SCRINSHOT we selected samples from all 12 regions that contained histological structures of interest, namely airways, alveoli and SMG. One of the tracheal samples (donor 3) was excluded from the analysis due to suboptimal mRNA quality of the sample. For Visium one representative sample from each anatomic region was processed.

6. In Sup fig 2 - no neutrophils were identified? This would be hard to believe for T cells?

We appreciate that the absence of some cell types is pointed out here.

Indeed, we could not identify neutrophils. Since no annotated neutrophils are present in scRNA-seq data which were used as a reference, they were initially absent from the probe set creation pipeline. The manually-added markers did not result in a formation of characteristic neutrophil cluster. Neutrophils have been detected in published diseased samples by scRNA-seq specially under inflammatory conditions [8], and their spatial identification in the inflamed samples should be possible.

T cells were sparsely detected by HybISS as we explain above. They were readily detected by the complementary SCRINSHOT experiments.

line 116, states that “additional cell type should not be assigned because of low abundance or sparse gene expression” - but is that a facet of the technique (ie the sensitivity) or really low levels of gene expression? And others have identified even rare cells such as tuft or ionocytes in lung tissue sections. How did this differ in each of the different locations - i.e. could you pick up tuft cells in the tracheal areas but not the proximal/distal lung regions?

We would like to clarify the context of line 116 “we manually annotated seven additional cell types that could not be assigned by the unsupervised sub-clustering of the HybISS data either due to their low abundance or sparse gene expression (T lymphocytes, NK cells, a mixed group of T and NK cells, ionocytes, tuft cells, rare tuft-like cells, squamous-like cells and aerocytes)”

This indicates either low RNA abundance (like for T-cells) or a technical limitation of the HybISS probe. Aerocytes were detected but did not cluster as a separate cluster because 2 of the 3 probes used for their detection was also expressed by other endothelial cells. The aerocytes are present dispersed in the endothelial cluster (Suppl. Fig. 2A,B). Squamous-like cells were also present in the suprabasal and epithelial “nan” clusters but did not cluster separately because they are too rare.

To complement this, we used SCRINSHOT panels on serial sections, and have also validated the presence of T cells using immunofluorescence. In the SCRINSHOT experiment we detected T cells by IL7R expression (up to 5% of all cell sin various tissue compartments). This was confirmed by T cell detection by immunofluorescence.

Immunofluorescent staining for CD4 (green) or CD8 (magenta), with the nuclei (DAPI, grey) in trachea (left), proximal airway and SMG (middle) and alveolar region (right). Scale bar 50 μ m.

Regarding the comment on rare epithelial cells. Indeed, ionocytes and tuft cells have been individually detected in previously published experiments. Our experiment aimed to localize them all together with other cell types, record their relative positions and abundances. In the HyBISS dataset, tuft cells were barely detectable in all airways, ionocytes were present in most tracheal samples, but not in smaller airways, whereas neuroendocrine cells formed two subclusters, *GRP* and *PCSK1N* positive, and were present throughout the airways:

Images of rare cells detected in HyBISS in samples from three anatomical regions with largest numbers of these cells (from web viewer).

To further investigate the rare cell types we designed a dedicated probe panel for SCRINSHOT to detect all rare cell types described in scRNA-seq data including tuft cells. We have detected tuft cells in all four patients based on the co-expression of *POU2F3* and *RGS13*. They were found in all three anatomical regions of the lung (Fig. 4D). We found that the only cell type with specific location preference was the *GHRL*-expressing neuroendocrine cells.

Bar plot of rare cell type frequencies across anatomical regions, error bars indicate SEM, n=2.

The sentence around line 116 now reads:

Based on positivity for corresponding cell type marker genes in the RNA-seq atlases [1-3], we manually annotated seven additional cell types that could not be assigned by the unsupervised sub-clustering of the HyBISS data either due to their sparsely detected gene expression, such as in T lymphocytes, NK cells, a group of T and/or NK cells (here labeled T/NK), and aerocytes, or due to their low abundance, such as ionocytes, tuft cells, rare tuft-like cells, and squamous-like cells (Suppl. Fig. 2B, Suppl. Table 5).

The immunofluorescence experiment for the two types NE is now added (Fig. 4C)

7. Much of the data for HyBISS and SCRINSHOT are presented together - but why - what does each signify? How are they comparable?

HyBISS was used to detect most of the cell types due its robustness and multiplexity, SCRINSHOT is more versatile and sensitive. It allows to detect specific cell states due to its high sensitivity. We have processed serial sections using these methods, making the results both comparable and complementary. At the same time one method is a validation of the other. HyBISS and SCRINSHOT results are presented together as they are both imaging-based and provide cellular resolution. We have also added statistical analysis correlating the results from these two methods (Suppl. Fig. 6B).

Line 122 - says that RRSST was used to "complement the HyBISS data - but why just HyBISS? Not SCRINSHOT? - What does this technique give that the others don't.

We apologise for the unclear description of the pipeline and the confusion it has caused. We have corrected the statement in line 122 and refined the description of the pipeline, as written in response to comment 2 above.

Visium was used to confirm both HyBISS and SCRINSHOT data (Suppl. Fig. 1C and Suppl. Fig. 7A-B) regarding the gene expression patterns and the cell type distribution. RRSST is a slightly modified version of Visium, suitable for difficult samples [3]. We used it in the proximal and distal lung regions to confirm the regional specificity of markers in the proximal and distal lung

regions and allow the identification of additional genes expressed in the same regions in an unbiased manner (Suppl. Fig. 9). We kept the Visium and RRST definitions and describe their in Methods and refer to the primary publications for details.

Supp fig 3 only shows proximal and distal lung- what about the trachea?

Thank you for this valid point. We have picked only two samples for the methods comparison, since trachea and proximal lung were generally composed of similar cell types, the only difference being their proportions. We therefore considered it sufficient to demonstrate only proximal and distal lung in the figures. Below we show the images of the trachea dataset with the three methods side-by-side (also available via the web viewer link): The upper row shows comparison of gene expression (*MUC5B*) and the lower annotation of cell types

Top: *MUC5B* expression in HybISS (normalized values by total transcript count per cell), SCRINSHOT (raw signal) and Visium (normalized values per spot) side by side with histological image (H&E). All three methods demonstrate *MUC5B* expression in the airway epithelium and submucosal gland. Bottom: The presence of cell types in all three methods (grouped to match spot clustering results of the Visium data).

8. Figure 2: The description of epithelial cell diversity and the methods for visualising cellular neighbourhoods and gene expression spatially are commendable. However, most of these representations show data from only one donor. The authors need to incorporate data from all donors, or a collective analysis of all donors, into the spatial distribution of genes and cell types? This may provide more informative insights.

We understand the concern regarding the data representation. The main figure shows one representative sample, where tissue images are used as examples. All analysis graphs are presented as a summary from all donors. We apologise that the summary of all four donors was not shown for baso-luminal gradient of gene expression. We have done the cumulative analysis for all detected genes in layers in all four donors in the tracheal epithelium, and added the description to the text in methods section:

In order to statistically define basal, luminal and intermediate-enriched genes, we have binned the distances into 10 groups and calculated the mean expression per distance. We used trachea as region with thickest epithelium. The top expression level (>1500 dots by sample) genes that appeared in all four donors were combined, and statistical comparison was performed to define layer differences using principal component analysis. Genes with lowest variance (<0.15) between donors were selected and tested for highest variability between layers (Std>0.05). The genes returned as a result were combined into a statistical plot.

We have added the statistical representation in the new Supplementary Figures 4 and 11. The plot of most variable genes that meet the selected statistical criteria is presented below. This defines characteristic basal, intermediate and luminal markers in all four donors (Figure 3E).

Box plot of distances to basal membrane from gene dots in tracheal epithelium. Values are accumulated by donor. Error bars indicate min and max values. Significant differences are calculated with nested one-way ANOVA followed by Tukey's multiple comparisons test, and indicated with lines over the brackets. Cell borders are ignored for this calculation.

9. 5.Line 155 - the authors have labelled cells as "nan -not annotated" – but its not clear why they are not annotated - they represent a relatively large proportion of cells in epithelium - was this true of all techniques? Still seem to have "nan" cells using several techniques as in supp fig 4?

We understand the important comment and explain our ‘nan’ definitions in detail below. The term ‘not annotated’ (nan) was used for cells that were either (i) negative for the distinctive cell type markers, or (ii) difficult to segment cells, positive for marker genes from more than one cell class (epithelial, immune, stromal, SMG, endothelial). For HybISS, where the gene panel was more extensive, the proportion of these cells was around 20% of all positive cells. In SCRINSHOT, where the cell type panel included less than half of marker genes, the proportion of nan cells was 20-40% depending on the location of the sample. Our interpretation of the existence of such cells in the tissue is one of the following: (i) the nuclear area expansion-based segmentation of the tissue was permitting signal mis-annotation from neighbor cells, or (ii) limitation in the extend of the probe panels. Here, HybISS has an advantage.

In addition to “nan”, we assigned not annotated cells within classes, such as “epithelial nan”, “immune nan” or “endothelial nan”. These groups include the cells distinctly expressing general cell class markers (*SLPI* for epithelial cells, *CD52* for immune cells, *RAMP2* for endothelial cells), but lacking specific cell type markers. Whether these cells represent new subclasses of cell types that were not captured in the previously published scRNA-seq based cell type annotations, or simply cells generally expressing low mRNA levels, remains to be investigated.

We added an additional paragraph commenting on the non-annotated cells:

However, a few of the expected cell types and subtypes could not be identified either due to the sparse signal from some probes, or due to the absence of particular cell types in the collected sections. Sparsity of signal, either from low gene expression, inefficient hybridization in thick tissue regions, or unsupervised probe selection, resulted not annotated cells (nan). In addition, nuclei expansion-based segmentation resulted in the appearance of doublets, that could not be assigned to a specific cell type, and were also labeled nan.

10. 6. Line 207 - conclusion for section - not clear what novelty is here?

The novelty is the presence of a gene expression program in the intermediate layers of the healthy stratified epithelium. This is in addition to the hillock like program which is occasionally detected and likely activated upon injury. We have re-written the last sentences of this section in order to highlight the novelty in gene expression patterns of the airway epithelium. The last paragraph includes the following statements:

*Our spatial analysis reveals multiple cell states located in layers of the pseudostratified tracheal epithelium. Overall, there is a strong correspondence to three layers along apical-basal axis of the airway epithelium, with characteristic *HSPB1* and *SERPINB3* expression in the intermediate layer, distinct from previously characterized hillock and squamous cell states [3, 7, 30], which in this study appear donor- and layer-variable (Suppl. Fig. 11H).*

11. 7. Fig 3 - section on rare cell types - I think this is potentially very interesting - but how rare is rare? The analysis of rare cell types is a true challenge with these tools.

Thank you for an interesting question, which needs to involve the comparison of multiple approaches to define such small cell populations. In the Human Lung Core Atlas [4] 107 donors were analysed, and among 207,023 airway epithelial cells 885 were annotated as rare (approximately 0.42%). In our study we analyzed 40,470 airway epithelial cells from 4 donors and annotated 156 rare cells (approximately 0.38%). The rare cell types include ionocytes, neuroendocrine and tuft cells, defined based on previous studies, including the adult scRNA-seq data (Sikkema et al). In the integrated scRNA-seq dataset the proportions of tuft cells were approximately 0.08%, for neuroendocrine cells 0.08%, and for ionocytes 0.27%. In our study these proportions are 0.06% (tuft), 0.24% (neuroendocrine) and 0.09% (ionocytes). The differences in these proportions could be related to the different number of probes and expression levels of markers used for each cell type.

The authors define four groups of neuroendocrine (NE) cells, though the expression of some markers, notably *CHGB* and *PCSK1N*, is very low (i.e. as low as two spots). Additional validation should be provided to confirm the existence of these cells. Ideally, this could be done through protein staining in tissue sections or using other datasets.

Publications that characterize NE cell subtypes were defined in embryonic studies [9, 10]. In the current study we characterize the NE states based on clusters by gene expression (image of clusters added to Suppl. Fig. 12A) among the rare cells. To define rare cells, we selected cells with positivity of rare cell marker genes (≥ 2 counts for lowly expressed genes, ≥ 5 counts for highly expressed genes). *ASCL3*, which was the only ionocyte marker. Then we clustered all positive cells based on all detected rare genes:

Clusters of rare cells from SCRINSHOT analysis (left) and neuroendocrine marker gene expression, normalized by total transcript count per cell (right), n=180.

These clusters include previously-defined categories: ionocytes, tuft or tuft-like cells, the GRP-positive and the *GHRL*-positive neuroendocrine cells. In addition, we detected a cluster expressing low *GRP* and no *GHRL*, but rather high levels of *ASCL1*, a canonical marker of neuroendocrine cells. An intermediate cell cluster with elevated levels of *PCSK1N* and *CHGB* co-expresses the other neuroendocrine markers to some degree. These markers were detected at variable levels, from very low (1-2 transcripts per cell for both genes) to intermediate (maximum 8 transcripts per cell for *PCSK1N*) and high (maximum 21 transcript per cell for *CHGB*). Cells with no transcripts for these genes also appeared among neuroendocrine cells. We report their presence location and interpret that these genes represent variations of the NE cell type (see figure below).

SCRINSHOT signal of rare cell type marker genes plotted on top of nuclei (DAPI, grey).

In conclusion, we cannot confirm that *PCSK1N*-high or *ASCL1*-high cells belong to a specific category of neuroendocrine cells, which may be reflected at the protein level, but we indicate that some of the markers of neuroendocrine cells have variable levels. This is depicted in the figure above (now moved to supplementary), which demonstrates mRNA detected at both low (only 2-3 dots per cell) and high (more than 10 dots per cell) levels. The groups of cells that we detect are based only on the list of the selected markers and cannot be treated as a complete list of all neuroendocrine cells, neither represents their full heterogeneity. We have picked the most abundant of the known NE markers. There are multiple other genes that can affect this variability.

It would be interesting to make a comparison to other more "common" cell types? For example in relation to basal cells, or goblet or ciliated cells. Overall, the data in fig 3 is difficult to follow.

- particularly 3C where the cells are very blurry – the reader needs a better description of what they are trying to depict here...

In order to demonstrate the existence of the two previously-characterised subtypes of embryonic NE cells in the adult lung, we have performed antibody staining for GHRL and GRP (Fig. 4C). We have also provided the new quantification of rare cell types per region in relation to all other cell types present in the airway epithelium (Fig. 4D).

Immunofluorescent staining for GHRL (green) and GRP (magenta) of two subtypes of neuroendocrine cells, as well as epithelial membrane marker CDH1 (white) on top of nuclei (DAPI, blue), distal lung of donor 4. Scale bar: 10 μ m.

We apologise for the low resolution in the primary pdf. We hope that the new version has images with higher resolution (see figure).

Why does the heatmap only represent two donors rather than all four?

We apologise for the unclear description. We clustered all four donors and find that cells from all four donors contribute to different clusters. Given the rarity of these cell types, this suggests that the cell types are found in four donors. UMAP plot was added to Suppl. Fig. 12A:

UMAP plot of rare cell clusters (left) and donors (right) from SCRINSHOT analysis.

12. 8. Fig 4 relates to the idea of different cell states existing in different tissue compartments, and talks about neighbourhood analysis (line 243) but I'm not clear where this analysis is presented or how it was done?

Thank you for this valuable comment. We have now added the paragraph in the Materials and Methods describing the histology-based tissue compartment assignment and cell state quantification:

Materials and Methods.

4.6. Analysis of cell states in histological tissue compartments

Histological tissue compartments were assigned manually on the tissue sections stained with hematoxylin and eosin after SCRINSHOT. The assigned regions were chosen to include the large airways, terminal and respiratory bronchioles, large and medium blood vessels, cartilage, submucosal gland and alveoli. The compartments included their surrounding connective tissue. (...)

The proportions of non-epithelial cells within these compartments were quantified according to their clusters and statistically compared between the anatomic location, donors and compartments using PERMANOVA statistical test (R). (...)

In order to make the analysis more understandable, we have re-arranged the order of the figures and chapters. In the new version the HybISS cell type assignment and neighborhood analysis is followed directly by the SCRINSHOT-based tissue compartment analysis.

Figure 4: The regions defined here are well described, and the spatial transcriptomics approach is used effectively to extract meaningful information. However, some aspects are confusing. Where is the data showing the different populations of endothelial cells expressing either SPARCL1 or CLDN5? These populations appear in the heatmap but were not shown earlier. Do they represent distinct clusters within the endothelial cells?

We greatly appreciate your insights on our results describing cell state locations. We are sorry that some of the aspects were confusing, and address this by adding information to the paper. The populations of endothelial cells appeared in clustering of SCRINSHOT dataset, which was not included in the initial figures. We have now presented this in supplementary Figure 5A-C.

UMAPs of cells from SCRINSHOT dataset from four donors (n=4 for lung, n=3 for trachea), coloured by (top) cell subtypes, and (bottom) gene expression of endothelial markers.

They seem to represent a small proportion of the data, and the grey-scale heatmap makes it difficult to visualise.

We apologize that the data in the heatmap is difficult to visualize.

We have replaced the greyscale heatmap with a coloured bar plot to improve viewing in Figure 2D. We note that *SPARCL1*-expressing endothelial cells are frequent in all compartments but *CLDN5*-expressing cells are more frequent in alveolar compartments.

Graph of mean cell type/subtype proportions in each of the histological compartments surrounding epithelium (peri-epithelial). Mean proportion per compartment is indicated in percent (%), n=3 in tracheal regions, n=4 in proximal and distal lung regions. Values were compared in lung regions using Friedman test followed by Dunn's multiple comparisons test. Significantly changing groups ($P < 0.05$) are indicated by asterisk (*). The cell subtypes that dominated in one anatomic location are highlighted with coloured boxes.

On what type of blood vessels are these cells located? The lung regions contain a vast network of capillaries, post-capillary venules, and small arterioles. It would be valuable to investigate where *SPARCL1*+ or *CLDN5*+ cells are located. The authors mention venous or capillary cells, but the images are too zoomed out to confirm these observations. Excellent antibodies are available for these molecules, and it would be crucial to stain lung sections to verify the presence of these markers on endothelial cells at the protein level.

In order to further address the question regarding the types of blood vessels, differentially expressing *SPARCL1* and *CLDN5*, we performed immunostaining. We have presented this in Suppl. Fig. 10.

The results of the staining indicate that the vessels around the airways are surrounded by a thin layer of ASMA (alpha smooth muscle actin), indicating that they belong to venule and/or arteriole category. The *CLDN5*-positive vessels located in the alveolar region lack ASMA staining, and belong to capillary network.

Donor 1

Donor 4

Immunofluorescent staining of distal lung samples for SPARCL1 (magenta), CLDN5 (green) and aSMA (blue). Nuclei are indicated by DAPI (white).

The summarizing statements were added to the text as follows:

In order to define the differences between SPARCL1 and CLDN5 expression in the vascular system, we first estimated the general distribution of endothelial subtypes in SCRINSHOT and HyBISS and defined that most of these cells are located in small vessels or capillaries and are of mixed origin (Suppl. Fig. 10A-B). Immunofluorescent staining confirmed that the vessels around the airways were surrounded by a thin layer of aSMA, indicating that they belong to venule and/or arteriole category, whereas CLDN5-positive vessels located in the alveolar region lacked aSMA staining, and belong to capillary network (Suppl. Fig. 10C-D). Therefore we concluded that the airways are surrounded by a network of small venules or arterioles, which are positive for SPARCL1 and low-positive for CLDN5. The cells expressing high levels of CLDN5 are most likely general (non-aerocyte) alveolar capillary cell population.

Most of the observations described in this section rely on heatmap representation, and the authors draw strong conclusions from this. Statistical analyses have been conducted, but the asterisks (*) are not clearly visible on the heatmap. Additionally, it is unclear which groups were compared. Was it individual donors or compartments within each donor? In the figure (fig 4) what are the n? Are they regions or patients or sections? If so how do they relate to the tissue sections? One from each or more? If so how were they selected?

We apologize that the description of statistical comparison was unclear. We would like to clarify that the statistical comparison of cell subtypes was performed as follows (added to the Material and Methods section 4.6):

For the statistical analysis of peri-epithelial compartments, the proportions of the non-epithelial cell types/states per compartment per region per donor were used. In case if multiple structures were present in the sample, the data was combined per donor. Therefore each cell type/state proportion in each of the peri-epithelial compartments gave one value per donor. Initially, the comparison of all compartments simultaneously was performed for each cell type using Friedman test followed by Dunn's multiple comparisons test. To complement that, the pairwise comparisons of these proportions were calculated for all cell types/states for each anatomical region separately, using the paired Wilcoxon ranked test. Kruskal-Wallis test was used for multiple compartment comparison for each cell type in non-epithelial compartments due to lack of donor matching per compartment.

The results of PERMANOVA indicated the differences between both compartments and donors. The post-test results of this analysis are combined in two graphs and all statistical differences are indicated with asterisk near the cell type title and a black line between groups in the peri-epithelial compartment graph above (Fig. 2D) as well as in non-epithelial compartment graph below (Suppl. Fig. 8E).

Graphs of mean cell type/subtype proportions in non-epithelial compartments. Mean proportion per compartment is indicated in percent. Values were compared in lung regions using Kruskal-Wallis test. Significantly changing groups ($P < 0.05$) are indicated by asterisk (*).

Friedmann's test followed by Dunn's post-test indicated that some cell types demonstrate significant variability between compartments. However, Wilcoxon ranked pairwise comparison test within each anatomical location did not give any significant results, when all cell types were compared simultaneously. We therefore can conclude that most of variability in subtype location is driven by anatomical location (tracheal, proximal, distal) rather than by the compartment within each location (peribronchial, alveolar, perivascular).

From the images, it appears that SPARCL1+ cells are close to T/NK cells. Are these T/NK cells different from the rest?

Thank you for the interesting observation. Indeed, it appears that these cells tend to be co-localised from the image that we used in the figure. Answering the question regarding the differences of these T/NK cells (including both T and NK cells) from the rest, is rather challenging. Unfortunately, we did not confidently detect the subtypes of T and NK cells in any dataset due to the markers used, and we therefore cannot answer this question directly. We are not aware of any robust markers that could separate the subtypes of NK cells. However different subtypes of T cells may have their preferential location. In order to answer the question about this group of immune cells in particular, we have performed immune staining for CD3, CD4, and CD8 markers to define the distribution of the specific T cell subtypes. The

majority of T cells in both alveolar and peribronchial locations were the CD4 cells, however CD8 cells were also widely present. The images of these stainings are presented below:

Immunofluorescent staining for major T cell markers demonstrating the general T cell population (left) positive for CD3 which is double-positive for either CD4 (yellow) or CD8 (magenta), staining surrounding the nuclei (DAPI blue, left, or grey, middle). Accumulations of T cells around the distal airway are shown on the right in two different donors, and are represented with both CD4 and CD8 cells. Scale bar 50 μ m.

Other areas distinct from distal peribronchial regions, which were found to have other types of endothelial cells, were also possessing scattered CD4 and CD8 positive cells:

Immunofluorescent staining for CD4 (green) or CD8 (magenta), with the nuclei (DAPI, grey) in trachea (left), proximal airway and SMG (middle) and alveolar region (right). Scale bar 50 μ m.

Therefore, we cannot observe any regional differences in the distribution of subtypes of T lymphocytes. They do share the location with SPARCL1-positive endothelial cells, however we do not believe that they are different from the rest of T cells based on the CD4 and CD8 markers.

What is the immune cell neighbourhood of the different endothelial cell populations?

We are grateful for the interesting question. In order to answer it, we utilized a separate spatial analysis, looking for cellular proximities within a certain range of endothelial cells. Firstly, it is

important to point out that the variations in the endothelial populations between anatomical regions included the domination of venous/capillary mixed population (*CLDN5*/*SPARCL1*/*IGFBP7*-triple positive in SCRINSHOT or *ACKR1*- or *FCN3*-positive in HybISS) endothelial cells in the trachea and *CLDN5*-high endothelial population in the distal lung compartments. Therefore we first compared immune and endothelial cell compositions between anatomical locations. No significant domination of any immune cell population was observed in the trachea and proximal lung, whereas in the distal lung we observed the presence of *APOE*-positive macrophages and an increase in *CLDN5* endothelial cells.

Bar plot of the proportions of immune and endothelial cells in each histological compartment.

Secondly, we performed a spatial co-localisation analysis, based on the close proximity (a distance of less than 23 μm from each endothelial cell) of different endothelial cell populations in each anatomical region. We calculated the proportions of all cell subtypes around endothelial cells, which included immune cells. Due to immune cells being highly variable between donors, no significant differences among the immune cell compositions of each endothelial cell subtype were identified.

Bar blot of mean percentage of immune cells in close proximity to each endothelial cell subtype in four donors. Error bars indicate standard deviation.

Immunofluorescent staining of tracheal, proximal and distal lung samples (donor 4) for SPARCL1 (red), CLDN5 (yellow) and aSMA (blue). Nuclei are indicated by DAPI (white). CLDN5 positive cells are evident in distal lung, most of them are double-positive for SPARCL1 and lack smooth muscle layer.

13. 8. Figure 5: The final part of the paper focuses on the analysis of COPD samples. Although the sample size is limited, the data presented are potentially interesting, and the methods and visualisations for analysing cellular neighbourhoods are excellent.

The methods section of the patient sample collection is confusing – for the healthy lung cell atlas it is stated that the tissue was taken from deceased organ donors – so these were all healthy, with no previous history of any respiratory conditions?

We have added clarifications in the methods section, stating *‘without any previously documented lung-related conditions’*. The healthy atlas part of the study was taken from donor organs that did not have any previous documented conditions that may be reflected in lung health. These organs were initially supposed to be transplanted, but were not for reasons other than organ condition. Yet, we rejected two out of six of such lungs due to pathological appearance only revealed in our histological analysis.

For the COPD and non-COPD samples – line 445, it doesn’t state how these patients were chosen? Tissue is taken from resections from cancer patients that had COPD - is this a good choice of tissue? Surely, they are unlikely to be normal at transcript level if from cancer patients, even if the histological assessment is “normal/cancerous”? In the table the donors are n=1, non-COPD and n=3, COPD - these are extremely small numbers to make definitive conclusions about anything.

The choice of diseased tissue was not aimed to characterize a pure disease phenotype, but rather to reveal the consistent and non-consistent alterations in each single sample to explore the utility of healthy atlas. The choice of samples was based on the availability of tissues with known pathologies. The control from cancer non-COPD patient provided the opportunity to detect additional variability, potentially caused by cancerous tissue in the surrounding regions. Indeed, transcriptionally peri-tumor tissue is likely to be different from the healthy lung. However, on the level of the cell type markers such changes were not visible. We demonstrate the differences not by pure statistical comparison of similar samples, but by revealing the outlier datasets (separated clusters) in process of data integration. The conclusion that we make is that atlas can be used to demonstrate both common (such as immune-enriched neighborhood) and sample-based (AT0- and TRB-enriched alveoli) changes, as each one of the samples stands out from the integrated data.

The AT0 cell population is characterised by the co-expression of NAPSA, SFTPC, SCGB3A1, SCGB3A2, and LTF. However, according to Supplementary Fig. 7C, some of these genes are not expressed by 100% of AT0 cells (e.g. LTF and SCGB3A1). How did the authors decide on these markers? Is the AT0 cell population heterogeneous? A visual representation of all AT cells in both healthy and COPD conditions would be helpful for readers.

The AT0 population was previously characterized as a cell state co-expressing alveolar type II and airway secretory markers [11]. In our study we have revealed that this population is heterogeneous, as defined in the healthy samples (line 196, Suppl. Fig. 5C and Fig. 3C). For the probe selection to detect them, we were guided by the previously-published information [11, 12] regarding the marker selection: the cells that possessed markers of both AT2 (*NAPSA*, *SFTPC* in our study) and airway secretory cells (*SCGB3A2*, *SCGB3A1*). The AT0 cells were separated from the general AT2 and airway secretory cells by subclustering, and we noticed that other secretory cell markers, like *LCN2* and *LTF*, are occasionally co-expressed in AT0 cells.

The expression of markers was shown in Suppl. Fig. 13D and the data were included in the viewer. To visualize AT0 gene expression in healthy and COPD samples here, include the co-expression of these markers in the images below.

Images of SCRINSHOT signal for characteristic AT0 markers. Scale bar 10 μ m.

The distribution of all AT cells in healthy and COPD conditions is presented in Figure 5B. In addition, we mapped AT0 and AT2 cells over histological images of healthy and COPD samples, as shown below. This panel has been added to Suppl. Fig. 14E.

Maps of AT0 (magenta) and AT2 (yellow) cells on top of histological images in healthy and COPD samples.

In conclusion, although AT0 cells are defined by the co-expression characteristic of AT2 cell markers *NAPSA* and/or *SFTPC* and airway secretory markers *SCGB3A2* and/or *SCGB3A1*, they occasionally express *LCN2* in healthy and *LTF* in COPD samples. The abundance of these cells is expected to be increased in the COPD samples, according to previous reports.

14. 9. The discussion is succinct and primarily summarises the study's limitations. It would benefit from additional discussion on the roles of rare neuroendocrine cells or specific endothelial cell populations, and their potential roles in lung function and COPD. Spatial transcriptomics has been successfully employed in other organs, and this could be discussed too.

I'm not clear what final conclusions were made from the data regarding our understanding of COPD?

We can only speculate about the role of revealed cell types. Yet we have added some points to the discussion. Regarding the COPD, the main conclusion is the overall shift towards AT0 and immune cell domination in the alveolar tissue replacing the healthy AT1 and capillary network. A future avenue of research may follow the altered immune neighborhood unveiled in the few patient samples that we analyzed here.

The points added to the text are:

- *We revealed a GHRL-expressing neuroendocrine cell group selectively in the distal airways, in contrast to their proximal location in embryonic lung development [32].*
- *The active role of AT0 cell state in other diseases [9] may be relevant to the repair processes in the early COPD stages.*
- *We detected CLDN5 expression in endothelial cells dominating in alveolar capillaries, whereas SPARCL1 being expressed in larger vessels.*

Minor

- The images in Figure 1D are too zoomed out, and higher magnification is needed to visualise the cellular level.

- Heatmaps are difficult to interpret in black and white; could they be changed to colour?
- Very difficult to see the yellow staining in Fig 2F - serpinb3 and SCGB1A1

We apologise for the picture quality. The issues have been addressed and the pictures have been modified accordingly.

Reviewer #2 (Remarks to the Author):

In the manuscript, Firsova et al. performed spatial transcriptomics (ST) profiling and analysis and presented two ST atlases of human lung: (1) a healthy lung atlas based on the ST profiling of samples from three representative regions of human lungs (n=4) using three technologies (HybISS, SCRINSHOT, and Visium/RRST) and (2) a COPD lung atlas based on SCRINSHOT profiling of distal lung samples from 3 patients with COPD GOLD stage II and 3 control lung samples. The authors first identified cell type marker gene panels from a previously published scRNA-seq of human lung and then used these panels for HybISS or SCRINSHOT-based ST profiling of healthy and COPD lung samples. Based on the HybISS data, 35 major cell types were identified in the healthy lung atlas. Based on the SCRINSHOT datasets, 23 major cell types were detected in the healthy lung, and 20 major cell types were defined in COPD. In addition to these major scRNA-seq-based cell types, the authors report the identification of intermediate cell states of suprabasal cells and SMG cells, as well as subtypes of neuroendocrine (NE) cells. In particular, they report the detection of GHRL-positive NE cells, a previously reported embryonic/pediatric NE subtype, in adult lungs and found that these cells were present in bronchioles but not in trachea or proximal lung. Using the healthy lung atlas, consistent regional variations in cell types and marker gene expression were identified. The patterns are consistent with existing knowledge and/or recent single-cell findings. The authors also integrated the ST data with cellular morphologies in H&E image and demonstrated the spatial patterns of specific cell types and marker gene expression in distinct tissue compartments.

Using the COPD atlas, changes in cell proportions are reported, including the decrease of AT1 cell proportions and the increase of T lymphocytes and AT0 cell proportions. The authors performed neighborhood analysis and identified three cellular neighborhoods enriched in COPD and two neighborhoods decreased in COPD. They proposed that these neighborhood enrichments and decreases were consistent or related to known COPD phenotypes or recent single cell findings, including the increase of AT0 cells with known inflammatory signatures of COPD.

Overall, the presented ST atlases of healthy and COPD lungs provide an initial resource for the research community for data exploration and understanding of the spatial patterns of marker genes and cellular neighborhoods in distinct tissue regions/compartments. The pipeline for tissue and bioinformatic analyses will be useful with the increasing interest in spatial transcriptomics for the study of the lung.

Major Concerns:

(1) Sample size and individual variations:

Only three COPD and three control lungs were included in the SCRINSHOT profiling of 41 genes for atlas construction and comparative analyses. Differences in age, sex, smoking, treatment regimen, and other patient characteristics limit conclusions for this small number of samples.

Within this small number of samples, COPD-related metadata was highly variant (Supp Table 1).

Regarding smoking status: Smoking status varied among COPD patients. Two of the three controls were non-smokers. Regarding sex: all three control patients were male, while two of the three COPD patients were female; one of the three controls was much younger than the COPD patients. Therefore, the small sample size, together with high individual variations in disease-related metadata, raises a major concern regarding the validity and utility of the comparative analysis.

We thank the reviewer for their support with constructive criticism and concrete suggestions. We acknowledge the concern regarding the sample size. A purpose with creating the healthy atlas was to provide a consistent healthy lung database and the methodology to integrate it with the spatial data from diseased lungs. The healthy lung samples were from males, female, smokers and non-smokers at different ages.

Our observations on the differences in the proportions of cell types and cell states between healthy and diseased samples are consistent with the ones previously detected in scRNA-seq data studies, containing larger cohorts of COPD samples from non-cancer patients (Rustam et al). This suggests that spatial analysis of a small number of patients with a limited panel of probes detects changes in cell type composition as larger datasets and additionally provides new information on potential, disease-related spatial relationships among the cell types. We also acknowledge the concern on metadata variability among the diseased samples. To address this we investigated whether the detected cell type variability relates to the patient metadata, or to the proportions of particular tissue structures and compartments (airways, alveolar tissue, and large blood vessels) in healthy and diseased samples (Supplementary Figure 14A-D).

Bar plots with individual data points indicating the proportions of each cell type within a particular tissue compartment in healthy and COPD samples. Error bars indicate SEM.

The plots above indicate that the variability within compartments is very limited, despite the metadata variability. In addition, the overall proportions and presence of cell types and classes are generally comparable between used samples.

(Left) Proportions of the indicated cell classes in the analyzed donors. (Right) UMAP plots of donors/patient identities from distal lung biopsies analyzed by SCRINSHOT. Groups of cells (arrows) indicate dominating contribution from COPD patients, whereas healthy samples are mixed.

(2) New insights into the disease:

Using the COPD atlas, the authors demonstrated changes in cell proportions and cellular neighborhoods in COPD, which were mostly consistent with or related to known COPD phenotypes in recent single cell RNA findings. It is not clear that new insights into disease pathogenesis have been generated from the present analyses.

A multiplex spatial cell type detection in situ in COPD samples has not been performed previously, and demonstrates a novel approach to characterize this disease. The construction of cellular neighborhoods in the COPD samples reveal a consistent immune-mesenchymal cell colocalisation, as well as an occasional expansion of the TRB zone. We also demonstrate that AT0 cells not only increase in numbers, but also occupy different niches in the distal lung, contributing to various neighborhoods. We would like to iterate that in this study we did not aim to explain the specificity or mechanisms of COPD progression, but rather demonstrate the utility of the healthy human lung cell map in the characterization of such a heterogeneous disease, which will require a future larger cohort and an extended panel of probes.

(3) AT0 cells in COPD lung:

Based on the analysis of the SCRINSHOT-based topographic lung atlas of COPD and healthy lungs, the present work reported a significant increase in the proportion of AT0 cells in the COPD samples.

According to the spatial visualization plot (Fig. 5B), the spatial locations of AT0 and AT2 cells look similar. Do AT0 and AT2 have similar or different spatial localization patterns in the COPD lung?

AT0 cells were defined as a population of cells co-expressing AT2 (NAPSA, SFTPC) and airway (SCGB3A1, SCGB3A2, LTF) markers. A recent scRNA-seq study (PMID: 35355018) identified an AT0 cell population co-expressing AT2 and airway markers, which were increased in COPD lungs. The study (PMID: 35355018) was not discussed in the “AT0 cell state alteration” section in the present work. What are the differences between the “AT0” in the present work and the one in PMID: 35355018?

Thank you for this insightful comment. In healthy datasets the AT0 cells were usually localized in the alveolar regions in proximity to large and small airways, and occasionally more distant, like classic AT2 cells. In two COPD samples AT0 cells were also observed near lymphatic structures, as well as elsewhere in alveolar tissue, indicating that they may be receiving signals from the surrounding tissue. However, the overall location pattern was not significantly changed in COPD. The highest increase in AT0 was found in the alveolar region (see Suppl. Fig. 14). We have added the description of location of AT0 vs AT2, as well as an extra figure demonstrating this, in the results section:

Histological compartment analysis indicated that these AT0 cells were mostly in the alveolar regions in proximity to the large and small airways, and in respiratory bronchioles, but the increase primarily affected alveolar regions (Suppl. Fig. 14A-E).

Maps of AT0 (magenta) and AT2 (yellow) cells on top of histological images in healthy and COPD samples.

In this context we apologise for missing the PMID 35355018 citation. This is an important paper with the description of AT0 occurrence in the healthy and diseased samples, where the name of this cell type was first mentioned. AT0 increase in COPD was also mentioned in the PMID 35355018, and could indeed be connected to some of the mechanisms demonstrated in the paper. We have added the citation and some discussion points to the text as follows:

Our relatively small experiment identified gene expression alterations including the increased proportion of AT0 and a relative reduction in AT1 cells in the diseased lungs, in line with previous knowledge of the COPD cell states and histopathology [9,34]. The active role of AT0 cell state in other diseases [9] may be relevant to the repair processes in the early COPD stages.

(4) GHRL-positive neuroendocrine (GHRL-NE) cells:

GHRL-NE cells were “detected in embryonic and pediatric datasets and were hypothesized to gradually disappear in adulthood” in a previous study. In the present work, GHRL-NE cells were identified in adult lungs and were found to be specifically located in distal bronchioles but not in trachea or proximal lung. Orthogonal experiments that support this conclusion would strengthen the work.

GHRL-positive neuroendocrine cells were additionally detected using immunofluorescence and presented in Figure 3C:

Immunofluorescent staining for GHRL (green) and GRP (magenta) of two subtypes of neuroendocrine cells, as well as epithelial membrane marker CDH1 (white) on top of nuclei (DAPI, blue), distal lung of donor 4. Scale bar: 10 μ m.

(5) Accuracy of cell type identification:

Cell types in both healthy and COPD lung atlases were identified mainly based on HybISS or SCRINSHOT profiling of pre-defined cell type marker gene panels, which were selected on the basis of published scRNA-seq datasets. These cell types formed the basis for major analyses in the present work, including cell type proportion analysis, identification of regional differences and variations, neighborhood analysis, and disease alteration analysis. Therefore, it is important to provide assessments of the accuracies of the identified cell types.

Thanks for this critical suggestion. Below we describe how we did the annotations and our new assessment of its congruence with previously published scRNA-seq data. We assigned 221,130 cells in HybISS and 164,719 cells in SCRINSHOT by hierarchical clustering. We first detected major groups of cells and identified large populations of SMG, epithelial, mesenchymal, alveolar epithelial and immune, and then subclustered each of the clusters to define cell types. The top 1-5 differentially expressed genes for each subcluster were used to define the cell types (Supplementary tables 4 for HybISS cell type annotation and 7 for SCRINSHOT cell type annotation). For some of the cell types that could not be identified, manual assignment by marker positivity was performed (Supplementary table 5. Criteria for selecting additional cell types in HybISS). This approach in cell type annotations was previously published for SCRINSHOT with the same probe panel [5].

The direct comparison of scRNA-seq and targeted spatial transcriptomics data can be challenging and inconclusive due limited probe numbers in targeted methods. In our study we have addressed this problem by combining three different SRT methods, one of which does not involve targeted probe binding, on serial section of the same samples (Visium). This approach allows drawing the conclusions for every targeted gene separately. We integrated the HybISS data, which are the basis of our annotations, with the scRNA-seq data of the corresponding tissue locations (Suppl. Fig 3A, also presented below).

In Supp Fig 2, dotplots were used to demonstrate the expression patterns of cell type marker genes in the published scRNA-seq cell types (panel C) and in the present HybISS cell types

(panel D). However, several markers did not exhibit a clear selective pattern in their corresponding HybISS cell types (Supp Fig. 2D), such as HPGD, secretory cell markers (*WFDC2*, *SCGB1A1*, *MUC5AC*), *APOE*, *DCN*, *MUC5B*, etc.

We apologise that Supplementary Figure 2 contains a plot that was difficult to interpret due to different scales of expression (normalized to the library size and scaled \log^{2+1} for scRNA-seq and true counts for HybISS). Potential scaling of the HybISS data would distort the data and we avoided to do it here. Additionally, cell types are shown with different granularities in the two graphs. The HybISS secretory cell population contains both club and goblet secretory subtypes, whereas in scRNA-seq it is split into subgroups, making the marker gene expression more localised.

A number of markers do not demonstrate the same relative strength of expression as in scRNA-seq data. We have added the description to the figure legend that this may be due to technical variability in probe performance. For example, secretory cell markers *WFDC2*, *SCGB1A1*, *MUC5AC* and *MUC5B* were detected at lower levels, compared to other secretory markers *BPIFB1* and *SLPI*, making their relative expression different. Since the HybISS data were not scaled, such low signal is not appearing in the plot. We present the raw mRNA signal for these genes below.

Spots indicating the decoded mRNA signal for the corresponding genes detected by HybISS on top of histological image of the bronchial epithelium (Proximal lung, Donor 3).

The genes that do not present a visible signal in the balloon plot in Supplementary Figure 2D, were indicated in brackets. These genes are expressed in HybISS-analysed samples, but the signal is weak and sparse. SCRINSHOT was more efficient in detecting strongly-expressed genes such as *SCGB1A1* (Suppl. Fig. 1C), *MUC5AC*, *MUC5B* and *APOE*. On the other hand, *DCN* expression was sparse in both SCRINSHOT and HybISS, compared to Visium. In the Visium data maximum *DCN* expression was detected efficiently in the thick connective tissue regions and around the cartilage, whereas in SCRINSHOT and HybISS only low background levels (one transcript per cell) appear rarely in epithelium and peribronchial parenchyma:

DCN expression signal detected by Visium (red color for highest detected signal), HybISS (green triangles indicate each detected transcript) and SCRINSHOT (green triangles indicate each detected transcript) projected on corresponding histological images.

Therefore, *DCN* detection in targeted methods could be affected by reduced probe binding in some tissue regions, such as the cartilage and thick connective tissue. These are the regions where very few transcripts were detected:

Left: all HybISS-detected transcripts (coloured shapes) plotted on top of nuclei (white). Right: histological image of the same section highlighting thick connective tissue areas.

Likewise, no expression of chondrocyte-enriched genes like *ACAN* and *CST3* were observed in the cartilage in SCRINSHOT and HybISS, yet enriched *ACAN*, *CST3*, *COL9A3* and *CYTL1* chondrocyte markers were observed in cartilage region in Visium and RRST. Efficient probe penetration and hybridization in dense tissues may require additional pre-treatment steps [13].

The detection of HPGD was a challenge with all three methods, it was both low abundant and rare, potentially due to the low number and morphology of the cells expressing it (aerocytes and mast cells).

HPGD expression signal in distal lung sample of donor 4 detected by Visium (red color for highest detected signal), HybISS and SCRINSHOT (red circles indicate each detected transcript) projected on corresponding tissue images (DAPI, white). The signal is overall low.

HPGD could not be used as a reliable marker, and other markers were used to define these cells.

Were all the annotated HybISS cell types supported by the selective and/or abundant expression of corresponding marker genes? It would be better if statistical analyses could be performed to quantify to support the accuracies of HybISS cell-type annotations.

We performed the suggested analysis, please see below and Suppl. Fig. 3A in the manuscript. We apologise that our analysis pipeline was not clearly described. For both HybISS and SCRINSHOT all the annotated cell types were supported by the abundant marker genes, according to the clustering analysis. Statistical analysis notebooks are available as supplementary information by this link: <https://github.com/alexandra-firsova/Human-lung-cell-atlas>.

In addition, we have applied the Mapping and Annotating Query Datasets method [14] for HybISS and SCRINSHOT to confirm that annotated clusters match the cell types annotated in the corresponding scRNA-seq dataset [15]. Below are the plots of the results of integration.

Balloon plot of HybISS cell types and their predicted annotations from scRNA-seq (level 2 classification of Madisson et al).

This plot indicates that the majority of cell types assigned in HybISS match the annotations of scRNA-seq according to the expression of 157 genes. Some of the less abundant and mixed/intermediate cell types do not show a strong correlation with scRNA-seq data due to their annotation by co-expression (no unique specific marker assigned, or a combination of markers used). For example, B lymphocytes were annotated by *CD79A* and *CD74*, co-expressed by B-plasma and AT2 and negatively to the B-plasma marker *JCHAIN*. In case of mucous cells, sparse gene expression of specific marker genes and shared expression with neighbor serous cells created a mixed prediction score. These cell types are labeled in brackets. The non-annotated cells or cells with low prediction scores were further compared spatially between the methods. These cells were found in their expected locations.

Maps of HybISS-annotated cell types on histological images

Cell types with low prediction scores mapped on histological tissue images of proximal lung sections. B lymphocytes and T/NK cells are mapped to the immune cell accumulation, endothelial nan cells appear both in arterial endothelial lining and sparsely in the tissue among capillaries, myofibroblasts are rare and located in proximity to smooth muscle, potentially being smooth muscle/fibroblast doublets. Pericytes are usually located in proximity to blood vessels, neuroendocrine cells - in the airway epithelium, myoepithelial cells – around submucosal gland. Mucous cells can be observed in the mucous tubules of submucosal gland, but occasionally appear in serous tubules, indicating that mucous annotation partially contains serous cells. SMG intermediate is mapped to both serous and mucous tubules of the submucosal gland, and indicates *PRR4*-negative population of serous and duct cells, but not exclusively SMG duct population.

This data was added to Suppl. Fig. 3, and the manuscript text was modified as follows:

The expected marker genes were expressed in the corresponding cell types, except for 11 genes, which were detected in very few cells of the annotated clusters (Suppl. Fig. 2C-D, in brackets). The overall performance of the HybISS marker gene probe panel and cell type annotations were tested using integration mapping [22] with the corresponding scRNA-seq dataset [1]. Cell types detected in HybISS demonstrated the highest prediction score to the expected scRNA-seq cell type annotations (Suppl. Fig. 3A). Ten cell types demonstrated prediction scores to more than one annotation, due to their mixed origin and marker co-expression with related types, or due to lack of cell type-specific markers. These cell types were annotated based on their location and morphology on histological images (Suppl. Fig. 3B). Their annotation and location was confirmed in complementary SCRINSHOT- and Visium-processed serial sections.

Likewise, visualization and quantification should be performed to support the accuracies of SCRINSHOT-based cell type annotations

For SCRINSHOT, which used several but more limited marker panels we show that assigned cell types express their expected marker genes in Supplementary Figure 4E-F. Further, we show

that the cell type annotations are consistent between SCRINSHOT, HybISS and Visium (Suppl. Fig. 7C)

Balloon plot with cell type marker genes for each assigned cell type in SCRINSHOT. Four of the cell types (Suprabasal intermediate, SMG intermediate, Myoepithelial and Schwann) did not have a specific marker gene in the panel, and were assigned by a combination of markers, as described in Suppl. Table 7).

Balloon plot with cell type marker genes used for SCRINSHOT cell type annotation, and their expression in cell types in scRNA-seq (Madisson et al).

Most cells possess a strong gene expression feature that allows them to group according to this feature and to form a cluster. Regarding the subtypes, these were mostly assigned by subclustering a parent cell type, and the cells in each subcluster showed at least one strong gene expression feature that allows them to group together according to previous knowledge.

(6) Robustness of cell type annotations and proportions:

In the healthy lung atlas, HyBISS and SCRINSHOT were applied to profile serial sections of the same region, providing an opportunity to assess the robustness of the reported cell type annotations and proportions. In line 128, the authors state that “the location of assigned cell types within the tissue was consistent between all three methods.” However, this conclusion is supported by visualization of subsets of cell type annotations in three regions (Supp Fig. 3). Statistical analyses should be performed to quantify and compare cell type annotations and cell type proportions between HyBISS and SCRINSHOT profiling of the same regions.

Thank you for a great suggestion. Since most serial sections cannot be perfectly aligned, statistical analysis of the full datasets would not reflect true overlap of cells. To circumvent this, we compared histological compartments in serial distal lung sections (each processed for H&E and Visium, or H&E and HyBISS or H&E and SCRINSHOT). Additionally, given the variability in the cell type classification granularity, we have combined some of the more specific types into larger groups (for example, endothelial, secretory cells and fibroblasts), creating a consistent cell type list between the three methods. This allowed a clearer combinatorial representation of the cell type distribution. The proportions of cell types within each compartment demonstrate significant correlation between the three methods in annotated histological compartments. The characteristic cell types driving the correlation are compartment-specific, and their proportions are similarly captured in the three methods.

These data have been added to Supplementary Figure 7C.

Detected cell type proportions within histological compartments

Pairwise correlation analysis of three methods by mapping cell type proportions within the same tissue compartments in distal lung. Left: selected areas analysed on three serial sections using three different methods. Right: XY pairwise correlation graphs of cell type proportions within each compartment with annotated cell types. P values are indicated next to r value as asterisks (* < 0.05, ** < 0.01, *** < 0.001, **** < 0.0001). Most abundant cell types of each compartment are labeled. Immune other (containing B plasma, dendritic cells and B lymphocytes) and macrophages are labeled blue due to their consistently higher levels detected in Visium, which can be explained by low levels of selected immune cell gene expression in HybISS and SCRINSHOT, and therefore some immune cells being beyond detection threshold.

Similarly, and following the reviewer's suggestions, we related the cell types identified by HybISS and SCRINSHOT in other anatomical regions. We selected three different histological compartments of proximal lung region of another donor. Again, the annotations are significantly correlated in each compartment. Data below and in suppl. Figure 6A-B.

Correlation analysis of two methods in serial sections from trachea (donor 2) and proximal lung (donor 3). (Top) Selection of the same histological compartments in HyBISS and SCRINSHOT in serial sections. (Bottom) Graphs with proportions of cell types correlated between SCRINSHOT (Y axis) and HyBISS (X axis), each dot representing a cell type. All analyzed cell types (13) are presented in all plots, cell types with highest abundance are labeled in each graph.

Overall, we conclude that the cell annotations are consistent for the 3 methods and reflect the cell-type description of the single cell mRNA sequencing data. In addition, our study extends these data by providing the locations and spatial relations of the cell types.

(7) Two of the control samples from the healthy lung atlas were also used in the COPD lung atlas and were re-profiled by SCRINSHOT using a gene panel (41 genes) different from the panel (64 genes) used for the healthy lung atlas providing an opportunity to cross-validate the cell type annotations and proportions in these two control samples, which could be important for identifying disease-related alterations.

Thank you for the suggestion. We must note that the sections used for COPD comparison were not direct serial sections from those used for the atlas, containing various proportions of area occupied by large tissue structures such as airways or blood vessels. However, we have added the direct comparison of the proportions of cell types detected by 41 and 64 genes to the Supplementary Figure 13.

The proportions of detected cell types from donors 2 and 4 from atlas (64 genes) in comparison with the COPD experiment (41 genes), in relation to total cell count (including the negative and not annotated cells), n=2.

This comparison indicates that 41 genes are sufficient to capture the 20 targeted cell types of the distal lung.

(8) Suprabasal vs. basal:

A detailed analysis of the spatial patterns of suprabasal cells was presented. *S100A2* was used as the primary marker to distinguish suprabasal from basal cells (lines 162 and 192). *S100A2* was selected based on the published scRNA-seq dataset. However, in the dotplot of Supp Fig. 2C, expression frequency and abundance in both basal and suprabasal cells in the scRNA-seq data was detected in *S100A2* shown in other bronchial epithelial cells.

How reliable and accurate is the separation of suprabasal and basal cells in the HyBISS data based on *S100A2* expression?

A first validation of markers *S100A2* and *KRT15* to distinguish basal/suprabasal cell layers was described in our previous publication [5]. We would like to clarify that *S100A2* (as well as *KRT5*) is not an exclusive marker for suprabasal cells, but it is co-expressed also by basal cells, whereas *KRT15* expression is missing from suprabasal layer. We have refined this analysis and demonstrated consistent differences of gene expression in the three layers of airway epithelium. This is presented in the figure below, which is included now in Figure 3E.

Box plot of distances to basal membrane from gene dots in tracheal epithelium. Values are accumulated by donor. Error bars indicate min and max values. Significant differences are calculated with nested one-way ANOVA followed by Tukey's multiple comparisons test, and indicated with lines over the brackets. Blue box indicates the genes that are located differently from both basal and luminal values. Cell borders are ignored for this calculation.

We have edited the text of the paper to clarify the layers, their markers, and the donor-variable gene expression:

Quantification of the mRNA signals along the distance from the basal membrane to the lumen defined basally-enriched (KRT15, IFITM1 and IFITM2), and apically-enriched mRNAs (BPIFB1,

CAPS), as well as an intermediately located gene expression program (SERPINB3 and HSPB1) [28] (Fig. 3E-F). In addition, S100A8 and SPRR1B, SPRR3 were variably expressed in different donors in scattered distances from the basal membrane (Suppl. Fig. 11), which could be related to epithelial condition [30].

Overall, there is a strong correspondence to three layers along apical-basal axis of the airway epithelium, with characteristic HSPB1 and SERPINB3 expression in the intermediate layer, distinct from previously characterized hillock and squamous cell states [3, 7, 31], which in this study appear donor- and layer-variable (Suppl. Fig. 11H).

(9) Data availability:

The ST data are shared through TissUUmaps-based web interfaces. We tried the web interfaces but found they were slow in loading data, for example, gene expression and metadata visualization.

It would be helpful if the data (such as the processed images and expression objects) could be shared for downloading for exploration and secondary data analysis.

All data information is currently available by the following link: <https://github.com/alexandra-firsova/Human-lung-cell-atlas>

Reviewer #3 (Remarks to the Author):

The manuscript describes a lung cell atlas that is spatially resolved using multiple SRT technologies including data from targeted and untargeted assay. The study revealed the locations of the majority of lung cell types including rare cell types, their variability across locations and distinct cellular neighborhoods, and as a demonstration, the usage of the atlas as a reference in the analysis of a COPD dataset. In general, the manuscript appears well written with substantial details and some novel observations.

I do have several friendly concerns:

1. The atlas is based on very small sample size (4 donors). The complete information about the samples (including age, gender, ethnicity, etc.) need to be disclosed and potential biases need to be discussed?

We thank the reviewer for the encouraging and constructive comments. We have added ethnicity to the sample metadata, which is available in Supplementary table 1. Since the focus of the current study was the gene expression and cell type variability in different regions of the same lungs, the limitation is the lack of statistical significance to demonstrate the donor variability due to small donor number. Despite the metadata variability (such as age, sex and smoking status), we observed consistent cell types, states and regional specific cellular neighborhoods in all of them. According to our observations, the variation in cell type proportions is less affected by patient metadata, but rather by the proportions of particular tissue structures and compartments (airways, alveolar tissue, and large blood vessels) in the sample.

We have added the notes about potential biases to discussion:

A limitation of our study is the lack of statistical analysis relating to the variability in age, sex and smoking status of the donors. Despite the limited donor number and including individuals with different physiological status, we observed consistent cell types, states and regional specific neighborhoods in all of them.

2. The study used 3 SRT technologies, including 2 targeted: SCRINSHOT and HyBISS as single-cell resolution, and 1 unbiased: Visium at spot resolution. Potential limitation/biases by the technologies need to be discussed and compatibility with other new (commercially available, and/or widely used) technologies such as Xenium, MERFISH, Stereo-seq, etc. needs to be clarified. E.g., how much overlap there is with the Xenium lung panel? How do others use the data from this study as a reference atlas when their data are produced by different technologies?

Thank you for the suggestion to discuss the SRT method compatibility. We have partially addressed this in Suppl. Figure 7, where we correlated proportions of detected cell types in three techniques. We would like to address the points of your question below.

1. There is compatibility between hybridization-based methods, however it is important to take into account the sensitivity and the size of the probe panel, and interpret the results accordingly.

2. The commercial Xenium lung panel consists of 289 genes with 39 gene overlap with HybISS panel and 14 genes used for SCRINSHOT. Since Xenium is a very similar padlock probe-based targeted method, the results for overlapping genes should be comparable, keeping in mind the variations in padlock probe number and concentration and therefore the strength of signal, which is unique for every experimental setup.

3. Other probe-based methods may be variable by the detection efficiency, however should be comparable in terms of spatial gene expression distribution and cellular neighborhoods.

In the supplementary data we provide the sequences of probes used for SCRINSHOT for the possibility to directly compare the results in own experimental setup.

The following phrases in the discussion are summarizing this:

A number of genes in both SCRINSHOT and HybISS showed weaker signal than expected. Yet there were strong marker genes in the designed panels that enabled the detection of most cell types, providing correlation between all three spatial methods.

The extent and open availability of our atlas for exploration and data mining together with the sensitivity and versatility of targeted methods suggest their further application in the analysis of cell type composition and gene expression changes in different regions of the healthy and diseased lungs.

3. How will the atlas be disseminated? It is great to have a web-based viewer? However, that is clearly insufficient, as it does not allow other users to integrate the entirety of the data as a reference into their studies.

The web-based viewer is accomplished for straightforward visualization purposes. However, the underlying data is available for download under the link: <https://github.com/alexandra-firsova/Human-lung-cell-atlas>

4. The data analysis performed looks largely reasonable (and clear) as described. However, to enable full reproducibility, the code/scripts used in generating the atlas and in performing the COPD analysis (e.g., Neighborhood enrichment analysis) need to be released publicly.

Html files with the code are available in supplementary data. We do not re-publish the code that was previously publishes and used in other publications.

Minor:

- Fig. 5A, can the asterisk for AT0, be moved near the “AT0” label to be consistent with other cell types?

Thank you for pointing this out. The asterisk has been moved as advised.

References

1. Gyllborg, D., et al., *Hybridization-based in situ sequencing (HybISS) for spatially resolved transcriptomics in human and mouse brain tissue*. *Nucleic Acids Res*, 2020. **48**(19): p. e112.
2. Sountoulidis, A., et al., *SCRINSHOT enables spatial mapping of cell states in tissue sections with single-cell resolution*. *PLoS Biol*, 2020. **18**(11): p. e3000675.
3. Mirzazadeh, R., et al., *Spatially resolved transcriptomic profiling of degraded and challenging fresh frozen samples*. *Nat Commun*, 2023. **14**(1): p. 509.
4. Sikkema, L., et al., *An integrated cell atlas of the lung in health and disease*. *Nat Med*, 2023. **29**(6): p. 1563-1577.
5. Kuemmerle, L.B., et al., *Probe set selection for targeted spatial transcriptomics*. *Nat Methods*, 2024. **21**(12): p. 2260-2270.
6. Li, J., et al., *Discrepant mRNA and Protein Expression in Immune Cells*. *Curr Genomics*, 2020. **21**(8): p. 560-563.
7. Kukanja, P., et al., *Cellular architecture of evolving neuroinflammatory lesions and multiple sclerosis pathology*. *Cell*, 2024. **187**(8): p. 1990-2009 e19.
8. Liao, M., et al., *Single-cell landscape of bronchoalveolar immune cells in patients with COVID-19*. *Nature Medicine*, 2020. **26**(6): p. 842-844.
9. He, P., et al., *A human fetal lung cell atlas uncovers proximal-distal gradients of differentiation and key regulators of epithelial fates*. *Cell*, 2022. **185**(25): p. 4841-4860 e25.
10. Sountoulidis, A., et al., *A topographic atlas defines developmental origins of cell heterogeneity in the human embryonic lung*. *Nat Cell Biol*, 2023.
11. Kadur Lakshminarasimha Murthy, P., et al., *Human distal lung maps and lineage hierarchies reveal a bipotent progenitor*. *Nature*, 2022. **604**(7904): p. 111-119.
12. Rustam, S., et al., *A Unique Cellular Organization of Human Distal Airways and Its Disarray in Chronic Obstructive Pulmonary Disease*. *Am J Respir Crit Care Med*, 2023. **207**(9): p. 1171-1182.
13. de Charleroy, C., A. Haseeb, and V. Lefebvre, *Preparation of Adult Mouse Skeletal Tissue Sections for RNA In Situ Hybridization*. *Methods Mol Biol*, 2021. **2245**: p. 85-92.
14. Hao, Y., et al., *Dictionary learning for integrative, multimodal and scalable single-cell analysis*. *Nature Biotechnology*, 2024. **42**(2): p. 293-304.
15. Madissoon, E., et al., *A spatially resolved atlas of the human lung characterizes a gland-associated immune niche*. *Nat Genet*, 2023. **55**(1): p. 66-77.

RESPONSES TO REVIEWERS' COMMENTS

Reviewer #1 (Remarks to the Author):

The authors have performed significant experiments to address my queries and have responded thoroughly to my comments.

However, it is important that the authors add a sentence at the beginning of the results section to state that of the four donors used for all of the subsequent analysis 3 were smokers. This really isn't clear from the text, and needs to be.

We greatly appreciate the reviewer's careful and extensive revision of our manuscript and the constructive feedback regarding the missing details. We have edited the sentence in the first paragraph of Results section as follows:

Samples from the remaining four donors, among which two were smokers, one ex-smoker and one non-smoker (Suppl. Table 1), were subjected to mRNA quality controls to reject the samples with low or diffuse RNA signal (Methods, Suppl. Fig. 1A).

Reviewer #2 (Remarks to the Author):

The authors have carefully addressed the specific concerns of each of the reviewers. They have clarified the methods, added the requested experimental data, described the limitations, and included appropriate references. All reviewers were concerned about the limited number and heterogeneity of the COPD and control subjects. This issue was not changed in the revision. The comparison, therefore, adds only modestly to current knowledge regarding anatomical or gene expression changes in COPD. Findings are consistent with known aspects of human lung structure and the pathological findings in COPD. The primary value of this work is methodological, providing a comparison of three spatial transcriptomics approaches. The study demonstrates useful cell markers and proximal-peripheral differences in cell types and gene expression patterns. In spite of patient and control heterogeneity, findings are generally consistent across all samples. Statistical approaches, probe sets, cell markers, and availability and accessibility of the dataset will be useful to the field.

In Summary:

Spatial transcriptomics methods and associated analytical frameworks are actively being developed by many laboratories, with the hope that they will yield novel mechanistic insights into the pathogenesis of lung disease.

We thank the reviewer for the careful evaluation of our revised manuscript and for the constructive comments. We also appreciate the recognition of the value of our findings. We acknowledge the concern regarding the limited number of samples. From the initial cohort of six control donors, we excluded two after identifying previously unreported

pathological alterations. This rigorous quality control ensured that the remaining four samples truly represented pathology-free tissue. Although these four donors differed in age, sex, and smoking history, our analyses focused on features that were consistently observed across all of them. Therefore, we believe that increasing the number of samples would not substantially alter the main conclusions, as the reported findings are those robust to inter-individual variability. While we agree that the sample size does not capture the full heterogeneity of stage II COPD, the primary aim of our study was to investigate COPD-associated changes at the single-cell and spatial neighborhood levels — an aspect not previously described.